# EVALUATING LOSSY COMPRESSION RATES OF DEEP GENERATIVE MODELS

## ABSTRACT

Deep generative models have achieved remarkable progress in recent years. Despite this progress, quantitative evaluation and comparison of generative models remains as one of the important challenges. One of the most popular metrics for evaluating generative models is the log-likelihood. While the direct computation of log-likelihood can be intractable, it has been recently shown that the log-likelihood of some of the most interesting generative models such as variational autoencoders (VAE) or generative adversarial networks (GAN) can be efficiently estimated using annealed importance sampling (AIS). In this work, we argue that the log-likelihood metric by itself cannot represent all the different performance characteristics of generative models, and propose to use rate distortion curves to evaluate and compare deep generative models. We show that we can approximate the entire rate distortion curve using one single run of AIS for roughly the same computational cost as a single log-likelihood estimate. We evaluate lossy compression rates of different deep generative models such as VAEs, GANs (and its variants) and adversarial autoencoders (AAE) on MNIST and CIFAR10, and arrive at a number of insights not obtainable from log-likelihoods alone.

## 1 INTRODUCTION

Generative models of images represent one of the most exciting areas of rapid progress of AI (Brock et al., 2019; Karras et al., 2018b;a). However, evaluating the performance of generative models remains a significant challenge. Many of the most successful models, most notably Generative Adversarial Networks (GANs) (Goodfellow et al., 2014), are *implicit generative models* for which computation of log-likelihoods is intractable or even undefined. Evaluation typically focuses on metrics such as the Inception score (Salimans et al., 2016) or the Fréchet Inception Distance (FID) score (Heusel et al., 2017), which do not have nearly the same degree of theoretical underpinning as likelihood-based metrics.

Log-likelihoods are one of the most important measures of generative models. Their utility is evidenced by the fact that likelihoods (or equivalent metrics such as perplexity or bits-per-dimension) are reported in nearly all cases where it's convenient to compute them. Unfortunately, computation of log-likelihoods for implicit generative models remains a difficult problem. Furthermore, log-likelihoods have important conceptual limitations. For continuous inputs in the image domain, the metric is often dominated by the fine-grained distribution over pixels rather than the high-level structure. For models with low-dimensional support, one needs to assign an observation model, such as (rather arbitrary) isotropic Gaussian noise (Wu et al., 2016). Lossless compression metrics for GANs often give absurdly large bits-per-dimension (e.g. $10^{14}$) which fails to reflect the true performance of the model (Grover et al., 2018; Danihelka et al., 2017). See Theis et al. (2015) for more discussion of limitations of likelihood-based evaluation.

Typically, one is not interested in describing the pixels of an image directly, and it would be sufficient to generate images close to the true data distribution in some metric such as Euclidean distance. For this reason, there has been much interest in Wasserstein distance as a criterion for generative models, since the measure exploits precisely this metric structure (Arjovsky et al., 2017; Gulrajani et al., 2017; Salimans et al., 2018). However, Wasserstein distance remains difficult to approximate, and hence it is not routinely used to evaluate generative models.

We aim to achieve the best of both worlds by measuring *lossy compression* rates of deep generative models. In particular, we aim to estimate the rate distortion function, which measures the number of bits required to match a distribution to within a given distortion. Like Wasserstein distance, it can exploit the metric structure of the observation space, but like log-likelihoods, it connects to the rich literature of probabilistic and information theoretic analysis of generative models. By focusing on different parts of the rate distortion curve, one can achieve different tradeoffs between the description length and the fidelity of reconstruction — thereby fixing the problem whereby lossless compression focuses on the details at the expense of high-level structure. It has the further advantage that the distortion metric need not have a probabilistic interpretation; hence, one is free to use more perceptually valid distortion metrics such as structural similarity (SSIM) (Wang et al., 2004) or distances between hidden representations of a convolutional network (Huang et al., 2018).

Algorithmically, computing rate distortion functions raises similar challenges to estimating log-likelihoods. We show that the rate distortion curve can be computed by finding the normalizing constants of a family of unnormalized probability distributions over the noise variables $\mathbf{z}$. Interestingly, when the distortion metric is squared error, these distributions correspond to the posterior distributions of $\mathbf{z}$ for Gaussian observation models with different variances; hence, the rate distortion analysis generalizes the evaluation of log-likelihoods with Gaussian observation models.

Annealed Importance Sampling (AIS) (Neal, 2001) is currently the most effective general-purpose method for estimating log-likelihoods of implicit generative models, and was used by Wu et al. (2016) to compare log-likelihoods of a variety of implicit generative models. The algorithm is based on gradually interpolating between a tractable initial distribution and an intractable target distribution. We show that when AIS is used to estimate log-likelihoods under a Gaussian observation model, the sequence of intermediate distributions corresponds to precisely the distributions needed to compute the rate distortion curve. Since AIS maintains a stochastic lower bound on the normalizing constants of these distributions, it automatically produces an upper bound on the *entire* rate distortion curve. Furthermore, the tightness of the bound can be validated on simulated data using bidirectional Monte Carlo (BDMC) (Grosse et al., 2015; Wu et al., 2016). Hence, we can approximate the entire rate distortion curve for roughly the same computational cost as a *single log-likelihood estimate*.

We use our rate distortion approximations to study a variety of variational autoencoders (VAEs) (Kingma & Welling, 2013), GANs and adversarial autoencoders (AAE) (Makhzani et al., 2015), and arrive at a number of insights not obtainable from log-likelihoods alone. For instance, we observe that VAEs and GANs have different rate distortion tradeoffs: While VAEs with larger code size can generally achieve better lossless compression rates, their performances drop at lossy compression in the low-rate regime. Conversely, expanding the capacity of GANs appears to bring substantial reductions in distortion at the high-rate regime without any corresponding deterioration in quality in the low-rate regime. We find that increasing the capacity of GANs by increasing the code size (width) has a qualitatively different effect on the rate distortion tradeoffs than increasing the depth. We also find that that different GAN variants with the same code size achieve nearly identical RD curves, and that the code size dominates the performance differences between GANs.

## 2 BACKGROUND

### 2.1 ANNEALED IMPORTANCE SAMPLING

Annealed importance sampling (AIS) (Neal, 2001) is a Monte Carlo algorithm based on constructing a sequence of $n$ intermediate distributions $p_k(\mathbf{z}) = \frac{\tilde{p}_k(\mathbf{z})}{Z_k}$, where $k \in \{0, \ldots, n\}$, between a tractable initial distribution $p_0(\mathbf{z})$ and the intractable target distribution $p_n(\mathbf{z})$. At the the $k$-th state ($0 \leq k \leq n$), the forward distribution $q_f$ and the un-normalized backward distribution $\tilde{q}_b$ are

$$q_f(\mathbf{z}_0, \ldots, \mathbf{z}_k) = p_0(\mathbf{z}_0)\mathcal{T}_0(\mathbf{z}_1|\mathbf{z}_0)\ldots\mathcal{T}_{k-1}(\mathbf{z}_k|\mathbf{z}_{k-1}), \tag{1}$$

$$\tilde{q}_b(\mathbf{z}_0, \ldots, \mathbf{z}_k) = \tilde{p}_k(\mathbf{z}_k)\tilde{\mathcal{T}}_{k-1}(\mathbf{z}_{k-1}|\mathbf{z}_k)\ldots\tilde{\mathcal{T}}_0(\mathbf{z}_0|\mathbf{z}_1), \tag{2}$$

where $\mathcal{T}_k$ is an MCMC kernel that leaves $p_k(\mathbf{z})$ invariant; and $\tilde{\mathcal{T}}_k$ is its reverse kernel. We run $M$ independent AIS chains, numbered $i = 1, \ldots, M$. Let $\mathbf{z}_k^i$ be the $k$-th state of the $i$-th chain. The

importance weights and normalized importance weights are

$$w_k^i = \frac{\tilde{q}_b(\mathbf{z}_1^i, \dots, \mathbf{z}_k^i)}{q_f(\mathbf{z}_1^i, \dots, \mathbf{z}_k^i)} = \frac{\tilde{p}_1(\mathbf{z}_1^i)}{p_0(\mathbf{z}_1^i)} \frac{\tilde{p}_2(\mathbf{z}_2^i)}{\tilde{p}_1(\mathbf{z}_2^i)} \cdots \frac{\tilde{p}_k(\mathbf{z}_k^i)}{\tilde{p}_{k-1}(\mathbf{z}_k^i)}, \qquad \tilde{w}_k^i = \frac{w_k^i}{\sum_{i=1}^M w_k^i}. \tag{3}$$

At the $k$-th step, the unbiased partition function estimate of $p_k(\mathbf{z})$ is $\hat{Z}_k = \frac{1}{M} \sum_{i=1}^M w_k^i$.

At the $k$-th step, we define the *AIS distribution* $q_k^{\text{AIS}}(\mathbf{z})$ as the distribution obtained by first sampling $\mathbf{z}_k^1, \dots, \mathbf{z}_k^M$ from the $M$ parallel chains using the forward distribution $q_f(\mathbf{z}_1^i, \dots, \mathbf{z}_M^i)$, and then re-sampling these samples based on $\tilde{w}_k^i$. More formally, the AIS distribution is defined as follows:

$$q_k^{\text{AIS}}(\mathbf{z}) = \mathbb{E}_{\prod_{i=1}^M q_f(\mathbf{z}_1^i, \dots, \mathbf{z}_k^i)} [\sum_{i=1}^M \tilde{w}_k^i \delta(\mathbf{z} - \mathbf{z}_k^i)]. \tag{4}$$

**Bidirectional Monte Carlo.** We know that the log partition function estimate $\log \hat{Z}$ is a *stochastic lower bound* on $\log Z$ (Jensen's inequality). As the result, using the forward AIS distribution as the proposal distribution results in a lower bound on the data log-likelihood. By running AIS in reverse, however, we obtain an upper bound on $\log Z$. However, in order to run the AIS in reverse, we need exact samples from the true posterior, which is only possible on the simulated data. The combination of the AIS lower bound and upper bound on the log partition function is called *bidirectional Monte Carlo* (BDMC), and the gap between these bounds is called the *BDMC gap* (Grosse et al., 2015). We note that AIS combined with BDMC has been used to estimate log-likelihoods for implicit generative models (Wu et al., 2016). In this work, we validate our AIS experiments by using the BDMC gap to measure the accuracy of our partition function estimators.

## 2.2 RATE DISTORTION THEORY

Let $\mathbf{x}$ be a random variable that comes from the data distribution $p_d(\mathbf{x})$. Shannon's fundamental compression theorem states that we can compress this random variable losslessly at the rate of $\mathcal{H}(\mathbf{x})$. But if we allow lossy compression, we can compress $\mathbf{x}$ at the rate of $R$, where $R \leq \mathcal{H}(\mathbf{x})$, using the code $\mathbf{z}$, and have a lossy reconstruction $\hat{\mathbf{x}} = f(\mathbf{z})$ with the distortion of $D$, given a distortion measure $d(\mathbf{x}, \hat{\mathbf{x}}) = d(\mathbf{x}, f(\mathbf{z}))$. The rate distortion theory quantifies the trade-off between the lossy compression rate $R$ and the distortion $D$. The rate distortion function $\mathcal{R}(D)$ is defined as the minimum number of bits per sample required to achieve lossy compression of the data such that the average distortion measured by the distortion function is less than $D$. Shannon's rate distortion theorem states that $\mathcal{R}(D)$ equals the minimum of the following optimization problem:

$$\min_{q(\mathbf{z}|\mathbf{x})} \mathcal{I}(\mathbf{z}; \mathbf{x}) \qquad s.t. \; \mathbb{E}_{q(\mathbf{x}, \mathbf{z})}[d(\mathbf{x}, f(\mathbf{z}))] \leq D. \tag{5}$$

where the optimization is over the *channel conditional* distribution $q(\mathbf{z}|\mathbf{x})$. Suppose the data-distribution is $p_d(\mathbf{x})$. The channel conditional $q(\mathbf{z}|\mathbf{x})$ induces the joint distribution $q(\mathbf{z}, \mathbf{x}) = p_d(\mathbf{x})q(\mathbf{z}|\mathbf{x})$, which defines the mutual information $\mathcal{I}(\mathbf{z}; \mathbf{x})$. $q(\mathbf{z})$ is the marginal distribution over $\mathbf{z}$ of the joint distribution $q(\mathbf{z}, \mathbf{x})$, and is called the *output marginal* distribution. We can rewrite the optimization of Eq. 5 using the method of Lagrange multipliers as follows:

$$\min_{q(\mathbf{z}|\mathbf{x})} \mathcal{I}(\mathbf{z}; \mathbf{x}) + \beta \mathbb{E}_{q(\mathbf{z}, \mathbf{x})}[d(\mathbf{x}, f(\mathbf{z}))]. \tag{6}$$

## 2.3 IMPLICIT GENERATIVE MODELS

The goal of generative modeling is to learn a model distribution $p(\mathbf{x})$ to approximate the data distribution $p_d(\mathbf{x})$. Implicit generative models define the model distribution $p(\mathbf{x})$ using a latent variable $\mathbf{z}$ with a fixed prior distribution $p(\mathbf{z})$ such as a Gaussian distribution, and a decoder or generator network which computes $\hat{\mathbf{x}} = f(\mathbf{z})$. In some cases (e.g. VAEs, AAEs), the generator explicitly parameterizes a conditional distribution $p(\mathbf{x}|\mathbf{z})$, such as a Gaussian observation model $\mathcal{N}(\mathbf{x}; f(\mathbf{z}), \sigma^2 \mathbf{I})$. But in implicit models such as GANs, the generator directly outputs $\hat{\mathbf{x}} = f(\mathbf{z})$. In order to treat VAEs and GANs under a consistent framework, we ignore the Gaussian observation model of VAEs (thereby treating the VAE decoder as an implicit model), and use the squared error distortion of $d(\mathbf{x}, f(\mathbf{z})) = \|\mathbf{x} - f(\mathbf{z})\|_2^2$. However, we note that it is also possible to assume a Gaussian observation model with a fixed $\sigma^2$ for GANs, and use the Gaussian negative log-likelihood (NLL) as the distortion measure for both VAEs and GANs: $d(\mathbf{x}, f(\mathbf{z})) = -\log \mathcal{N}(\mathbf{x}; f(\mathbf{z}), \sigma^2 \mathbf{I})$. This is equivalent to squared error distortion up to a linear transformation.

## 3 RATE-PRIOR DISTORTION FUNCTIONS

In this section, we describe the *rate-prior distortion function*, as a variational upper bound on the true rate distortion function.

### 3.1 VARIATIONAL BOUNDS ON MUTUAL INFORMATION

We must modify the standard rate distortion formalism slightly in order to match the goals of generative model evaluation. Specifically, we are interested in evaluating lossy compression with coding schemes corresponding to *particular* trained generative models, including the fixed prior $p(\mathbf{z})$. For models such as VAEs, $\mathrm{KL}(q(\mathbf{z}|\mathbf{x})\|p(\mathbf{z}))$ is standardly interpreted as the description length of $\mathbf{z}$. Hence, we adjust the rate distortion formalism to use $\mathbb{E}_{p_d(\mathbf{x})}\mathrm{KL}(q(\mathbf{z}|\mathbf{x})\|p(\mathbf{z}))$ in place of $\mathcal{I}(\mathbf{x}, \mathbf{z})$, and refer to this as the *rate-prior objective*. The rate-prior objective upper bounds the standard rate:

$$\mathcal{I}(\mathbf{x}; \mathbf{z}) \leq \mathcal{I}(\mathbf{x}; \mathbf{z}) + \mathrm{KL}(q(\mathbf{z})\|p(\mathbf{z})) = \mathbb{E}_{p_d(\mathbf{x})}\mathrm{KL}(q(\mathbf{z}|\mathbf{x})\|p(\mathbf{z})). \quad (7)$$

In the context of variational inference, $q(\mathbf{z}|\mathbf{x})$ is the posterior, $q(\mathbf{z}) = \int p_d(\mathbf{x})q(\mathbf{z}|\mathbf{x})d\mathbf{x}$ is the aggregated posterior (Makhzani et al., 2015), and $p(\mathbf{z})$ is the prior. In the context of rate distortion theory, $q(\mathbf{z}|\mathbf{x})$ is the channel conditional, $q(\mathbf{z})$ is the output marginal, and $p(\mathbf{z})$ is the *variational output marginal* distribution. The inequality is tight when $p(\mathbf{z}) = q(\mathbf{z})$, i.e., the variational output marginal (prior) is equal to the output marginal (aggregated posterior). We note that the upper bound of Eq. 7 has been used in other algorithms such as the Blahut-Arimoto algorithm (Arimoto, 1972) or the variational information bottleneck algorithm (Alemi et al., 2016).

### 3.2 RATE-PRIOR DISTORTION FUNCTIONS

Analogously to the rate distortion function, we define the *rate-prior distortion* function $\mathcal{R}_p(D)$ as the minimum value of the rate-prior objective for a given distortion $D$. More precisely, $\mathcal{R}_p(D)$ is the solution of

$$\min_{q(\mathbf{z}|\mathbf{x})} \mathbb{E}_{p_d(\mathbf{x})}\mathrm{KL}(q(\mathbf{z}|\mathbf{x})\|p(\mathbf{z})) \qquad s.t. \ \mathbb{E}_{q(\mathbf{x},\mathbf{z})}[d(\mathbf{x}, f(\mathbf{z}))] \leq D. \quad (8)$$

We can rewrite the optimization of Eq. 8 using the method of Lagrange multipliers as follows:

$$\min_{q(\mathbf{z}|\mathbf{x})} \mathbb{E}_{p_d(\mathbf{x})}\mathrm{KL}(q(\mathbf{z}|\mathbf{x})\|p(\mathbf{z})) + \beta\mathbb{E}_{q(\mathbf{x},\mathbf{z})}[d(\mathbf{x}, f(\mathbf{z}))]. \quad (9)$$

Conveniently, the Lagrangian decomposes into independent optimization problems for each $\mathbf{x}$, allowing us to treat this as an optimization problem over $q(\mathbf{z}|\mathbf{x})$ for fixed $\mathbf{x}$. We can compute the rate distortion curve by sweeping over $\beta$ rather than by sweeping over $D$.

Now we describe some of the properties of the rate-prior distortion function $\mathcal{R}_p(D)$, which are straightforward analogues of well-known properties of the rate distortion function.

**Proposition 1.** $\mathcal{R}_p(D)$ has the following properties:

(a) $\mathcal{R}_p(D)$ is non-increasing and convex function of $D$.

(b) We have $\mathcal{R}(D) = \min_{p(\mathbf{z})} \mathcal{R}_p(D)$. As a corollary, for any $p(\mathbf{z})$, we have $\mathcal{R}(D) \leq \mathcal{R}_p(D)$.

(c) The rate-prior distortion optimization of Eq. 9 has a unique global optimum which can be expressed as $q_\beta^*(\mathbf{z}|\mathbf{x}) = \frac{1}{Z_\beta(\mathbf{x})}p(\mathbf{z})\exp(-\beta d(\mathbf{x}, f(\mathbf{z})))$.

*Proof.* The proofs are provided in Appendix C.1.

Prop. 1b states that for any prior $p(\mathbf{z})$, $\mathcal{R}_p(D)$ is a variational upper-bound on $\mathcal{R}(D)$. More specifically, we have $\mathcal{R}(D) = \min_{p(\mathbf{z})} \mathcal{R}(D)$, which implies that for any given $\beta$, there exists a prior $p_\beta^*(\mathbf{z})$, for which the variational gap between rate distortion and rate-prior distortion functions at $\beta$ is zero. Fig. 1a shows a geometrical illustration of Prop. 1b for three values of $\beta \in \{0.25, 1, 4\}$. We can see in this figure that all $\mathcal{R}_p(D)$ curves are upper bounds on $\mathcal{R}(D)$, and for any given $\beta$, $\mathcal{R}_{p_\beta^*}(D)$ is tangent to both $\mathcal{R}_p(D)$ and to the line with the slope of $\beta$ passing through the optimal solution.

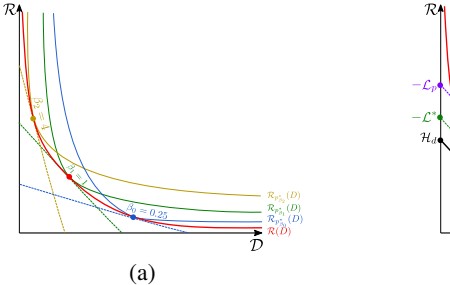 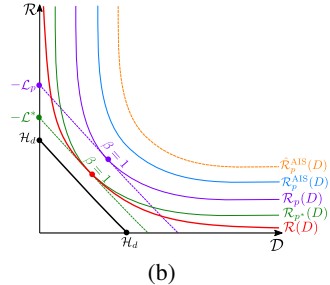

(a)             (b)

Figure 1: The rate-prior distortion function with (a) arbitrary distortion function $d(\mathbf{x}, f(\mathbf{z}))$ and (b) negative log-likelihood distortion function of $-\log p(\mathbf{x}|\mathbf{z})$.

### 3.3 RATE-PRIOR DISTORTION FUNCTIONS WITH NLL DISTORTION

If the decoder outputs a probability distribution (as in a VAE), we can define the distortion metric to coincide with the negative log-likelihood (NLL): $d(\mathbf{x}, f(\mathbf{z})) = -\log p(\mathbf{x}|\mathbf{z})$. We now describe some of the properties of the rate-prior distortion functions with NLL distortions.

**Proposition 2.** The rate-prior distortion function $\mathcal{R}_p(D)$ with NLL distortion of $-\log p(\mathbf{x}|\mathbf{z})$ has the following properties:

    (a) $\mathcal{R}(D)$ is lower bounded by the linear function of $\mathcal{H}_d - D$, and upper bounded by the rate-prior distortion function: $\mathcal{H}_d - D \leq \mathcal{R}(D) \leq \mathcal{R}_p(D)$.

    (b) The global optimum of rate-prior distortion optimization (Eq. 9) can be expressed as $q_\beta^*(\mathbf{z}|\mathbf{x}) = \frac{1}{Z_\beta^*(\mathbf{x})} p(\mathbf{z})p(\mathbf{x}|\mathbf{z})^\beta$, where $Z_\beta^*(\mathbf{x}) = \int p(\mathbf{z})p(\mathbf{x}|\mathbf{z})^\beta d\mathbf{z}$.

    (c) At $\beta = 1$, the negative summation of rate-prior and distortion is the true log-likelihood: $\mathcal{L}_p = \mathbb{E}_{p_d(\mathbf{x})}[\log p(\mathbf{x})] = -R_\beta\big|_{\beta=1} - D_\beta\big|_{\beta=1}$.

*Proof.* The proofs are provided in Appendix C.2.

Fig. 1b shows the geometrical illustration of Prop. 2. We can see that according to Prop. 2a the rate distortion function $\mathcal{R}(D)$ is sandwiched between the linear function $\mathcal{H}_d - D$ and the rate-prior distortion $\mathcal{R}_p(D)$. At $\beta = 1$, let $\mathcal{L}^*$ and $\mathcal{L}_p$ be the negative summation of rate and distortion on the rate distortion and rate-prior distortion curves respectively (shown in Fig. 1b). From Prop. 2c we know that $\mathcal{L}_p$ is the true log-likelihood of the generative model. From Prop. 1b, we can conclude that $\mathcal{L}^* = \max_{p(\mathbf{z})} \mathcal{L}_p$. This reveals an important relationship between rate distortion theory and generative modeling that was also observed in Lastras (2019): for a given generative model with a fixed conditional $p(\mathbf{x}|\mathbf{z})$, the best log-likelihood $\mathcal{L}_p$ that can be achieved by optimizing the prior $p(\mathbf{z})$ is the $\mathcal{L}^*$, which can be found by solving the rate distortion problem. Furthermore, the corresponding optimal prior $p^*(\mathbf{z})$ is the output marginal of the optimal channel conditional of the rate distortion problem at $\beta = 1$. Fig. 1b shows the rate-prior distortion function $\mathcal{R}_{p^*}(D)$ corresponding to $p^*(\mathbf{z})$. In a "good" generative model, where the model distribution is close to the data-distribution, the negative log-likelihood $-\mathcal{L}_p$ is close to the entropy of data $\mathcal{H}_d$, and the variational gap between $\mathcal{R}_p(D)$ and $\mathcal{R}(D)$ is tight.

## 4 BOUNDING RATE-PRIOR DISTORTION FUNCTIONS WITH AIS

In the previous section, we introduced the rate-prior distortion function $\mathcal{R}_p(D)$ and showed that it upper bounds the true rate distortion function $\mathcal{R}(D)$. However, evaluating $\mathcal{R}_p(D)$ is also intractable. In this section, we show how we can upper bound $\mathcal{R}_p(D)$ using a single run of the AIS algorithm.

**AIS Chain.** We fix a temperature schedule $0 = \beta_0 < \beta_1 < \ldots < \beta_n = \infty$. For the $k$-th intermediate distribution, we use the optimal channel conditional $q_k(\mathbf{z}|\mathbf{x})$ and partition function $Z_k(\mathbf{x})$, corresponding to points along $\mathcal{R}_p(D)$ and derived in Prop. 1c:

$$q_k(\mathbf{z}|\mathbf{x}) = \frac{1}{Z_k}\tilde{q}_k(\mathbf{z}|\mathbf{x}), \quad \text{where} \quad \tilde{q}_k(\mathbf{z}|\mathbf{x}) = p(\mathbf{z})\exp(-\beta_k d(\mathbf{x}, f(\mathbf{z}))), \quad Z_k(\mathbf{x}) = \int \tilde{q}_k(\mathbf{z}|\mathbf{x})d\mathbf{z}. \quad (10)$$

Conveniently, this choice coincides with geometric averages, the typical choice of intermediate distributions for AIS, i.e, the $k^{th}$ step happens to be the optimal solutions for $\beta_k$. This chain is shown in Fig. 2. For the transition operator, we use Hamiltonian Monte Carlo (Neal et al., 2011). At the $k$-th step, the rate-prior $R_k(\mathbf{x})$ and the distortion $D_k(\mathbf{x})$ are

$$R_k(\mathbf{x}) = \mathrm{KL}(q_k(\mathbf{z}|\mathbf{x})\|p(\mathbf{z})), \qquad D_k(\mathbf{x}) = \mathbb{E}_{q_k(\mathbf{z}|\mathbf{x})}[d(\mathbf{x}, f(\mathbf{z}))]. \tag{11}$$

**AIS Rate-Prior Distortion Curve.** For each data point $\mathbf{x}$, we run $M$ independent AIS chains, numbered $i = 1, \ldots, M$, in the forward direction. At the $k$-th state of the $i$-th chain, let $\mathbf{z}_k^i$ be the state, $w_k^i$ be the AIS importance weights, and $\tilde{w}_k^i$ be the normalized AIS importance weights. We denote the AIS distribution at the $k$-th step as the distribution obtained by first sampling from all the $M$ forward distributions $q_f(\mathbf{z}_1^i, \ldots, \mathbf{z}_k^i|\mathbf{x})\big|_{i=1:M}$, and then re-sampling the samples based on their normalized importance weights $\tilde{w}_k^i$ (see Section 2.1 and Appendix C.4 for more details). More formally $q_k^{\mathrm{AIS}}(\mathbf{z}|\mathbf{x})$ is

$$q_k^{\mathrm{AIS}}(\mathbf{z}|\mathbf{x}) = \mathbb{E}_{\prod_{i=1}^M q_f(\mathbf{z}_1^i, \ldots, \mathbf{z}_k^i|\mathbf{x})}\Big[\sum_{i=1}^M \tilde{w}_k^i \delta(\mathbf{z} - \mathbf{z}_k^i)\Big]. \tag{12}$$

Using the AIS distribution $q_k^{\mathrm{AIS}}(\mathbf{z}|\mathbf{x})$ defined in Eq. 12, we now define the AIS distortion $D_k^{\mathrm{AIS}}(\mathbf{x})$ and the AIS rate-prior $R_k^{\mathrm{AIS}}(\mathbf{x})$ as follows:

$$D_k^{\mathrm{AIS}}(\mathbf{x}) = \mathbb{E}_{q_k^{\mathrm{AIS}}(\mathbf{z}|\mathbf{x})}[d(\mathbf{x}, f(\mathbf{z}))] \qquad R_k^{\mathrm{AIS}}(\mathbf{x}) = \mathrm{KL}(q_k^{\mathrm{AIS}}(\mathbf{z}|\mathbf{x})\|p(\mathbf{z})) \tag{13}$$

We now define the *AIS rate-prior distortion curve* $\mathcal{R}_p^{\mathrm{AIS}}(D)$ (shown in Fig. 1b) as the RD cruve obtained by tracing pairs of $\big(R_k^{\mathrm{AIS}}(\mathbf{x}), D_k^{\mathrm{AIS}}(\mathbf{x})\big)$.

**Proposition 3.** The AIS rate-prior distortion curve upper bounds the rate-prior distortion function: $\mathcal{R}_p^{\mathrm{AIS}}(D) \geq \mathcal{R}_p(D)$.

*Proof.* The proof is provided in Appendix C.4.

**Estimated AIS Rate-Prior Distortion Curve.** Although the AIS distribution can be easily sampled from, its density is intractable to evaluate. As the result, evaluating $\mathcal{R}_p^{\mathrm{AIS}}(D)$ is also intractable. We now propose to evaluate an upper-bound on $\mathcal{R}_p^{\mathrm{AIS}}(D)$ by finding an upper bound for $R_k^{\mathrm{AIS}}(\mathbf{x})$, and an unbiased estimate for $D_k^{\mathrm{AIS}}(\mathbf{x})$. We use the AIS distribution samples $\mathbf{z}_k^i$ and their corresponding weights $\tilde{w}_k^i$ to obtain the following distortion and partition function estimates:

$$\hat{D}_k^{\mathrm{AIS}}(\mathbf{x}) = \sum_i \tilde{w}_k^i d(\mathbf{x}, f(\mathbf{z}_k^i))), \qquad \hat{Z}_k^{\mathrm{AIS}}(\mathbf{x}) = \frac{1}{M}\sum_i w_k^i. \tag{14}$$

Having found the estimates $\hat{D}_k^{\mathrm{AIS}}(\mathbf{x})$ and $\hat{Z}_k^{\mathrm{AIS}}(\mathbf{x})$, we propose to estimate the rate as follows:

$$\hat{R}_k^{\mathrm{AIS}}(\mathbf{x}) = -\log \hat{Z}_k^{\mathrm{AIS}}(\mathbf{x}) - \beta_k \hat{D}_k^{\mathrm{AIS}}(\mathbf{x}). \tag{15}$$

We define the *estimated AIS rate-prior distortion curve* $\hat{\mathcal{R}}_p^{\mathrm{AIS}}(D)$ (shown in Fig. 1b) as an RD curve obtained by tracing pairs of rate distortion estimates $\big(\hat{R}_k^{\mathrm{AIS}}(\mathbf{x}), \hat{D}_k^{\mathrm{AIS}}(\mathbf{x})\big)$.

**Proposition 4.** The estimated AIS rate-prior distortion curve upper bounds the AIS rate-prior distortion curve in expectation: $\mathbb{E}[\hat{\mathcal{R}}_p^{\mathrm{AIS}}(D)] \geq \mathcal{R}_p^{\mathrm{AIS}}(D)$. More specifically, we have

$$\mathbb{E}[\hat{R}_k^{\mathrm{AIS}}(\mathbf{x})] \geq R_k^{\mathrm{AIS}}(\mathbf{x}), \qquad \mathbb{E}[\hat{D}_k^{\mathrm{AIS}}(\mathbf{x})] = D_k^{\mathrm{AIS}}(\mathbf{x}). \tag{16}$$

*Proof.* The proof is provided in Appendix C.4.

In summary, from Prop. 1, Prop. 3 and Prop. 4, we can conclude that the estimated AIS rate-prior distortion curve upper bounds the true rate distortion curve in expectation (shown in Fig. 1b):

$$\mathbb{E}[\hat{\mathcal{R}}_p^{\mathrm{AIS}}(D)] \geq \mathcal{R}_p^{\mathrm{AIS}}(D) \geq \mathcal{R}_p(D) \geq \mathcal{R}(D). \tag{17}$$

In all the experiments, we plot the estimated AIS rate-prior distortion function $\hat{\mathcal{R}}_p^{\mathrm{AIS}}(D)$.

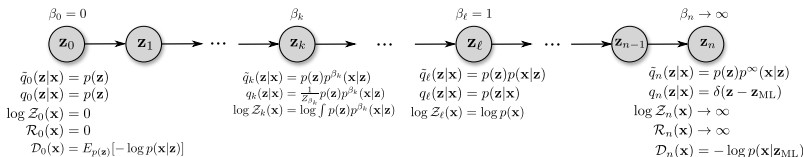

Figure 2: AIS chain for estimating the rate-prior distortion function with NLL distortion.

**Accuracy of AIS Estimates.** While the above discussion focuses on obtaining upper bounds, we note that AIS is one of the most accurate general-purpose methods for estimating partition functions, and therefore we believe our AIS upper bounds to be fairly tight in practice. In theory, for large number of intermediate distributions, the AIS variance is proportional to $1/MK$ (Neal, 2001; 2005), where $M$ is the number of AIS chains and $K$ is the number of intermediate distributions. For the main experiments of our paper, we evaluate the tightness of the AIS estimate by computing the BDMC gap, and show that in practice our upper bounds are tight (Appendix D).

**The Rate Distortion Tradeoff in the AIS Chain.** Different values of $\beta$ corresponds to different tradeoffs between the compression rate and the distortion in a given generative model. $\beta = 0$ corresponds to the case where $q_0(\mathbf{z}|\mathbf{x}) = p(\mathbf{z})$. In this case, the compression rate is zero, and the distortion would be large, since in order to reconstruct $\mathbf{x}$, we simply sample from the prior and generate a random $\hat{\mathbf{x}}$ that is completely independent of $\mathbf{x}$. In this case, the distortion would be $\mathcal{D}_0(\mathbf{x}) = \mathbb{E}_{p(\mathbf{z})}[d(\mathbf{x}, f(\mathbf{z}))]$. In the case of probabilistic decoders with NLL distortion, another interesting intermediate distribution is $\beta_\ell = 1$, where the optimal channel conditional is the true posterior of the generative model $q_\ell(\mathbf{z}|\mathbf{x}) = p(\mathbf{z}|\mathbf{x})$. In this case, as shown in Prop. 2c, the summation of the rate-prior and the distortion term is the negative of true log-likelihood of the generative model. As $\beta_n \to \infty$, the network cares more about the distortion and less about the compression rate. In this case, the optimal channel conditional would be $q_n(\mathbf{z}|\mathbf{x}) = \delta(\mathbf{z} - \mathbf{z}_{\mathrm{ML}}(\mathbf{x}))$, where $\mathbf{z}_{\mathrm{ML}}(\mathbf{x}) = \arg\min_{\mathbf{z}} d(\mathbf{x}, f(\mathbf{z}))$. In other words, since the network only cares about the distortion, the optimal channel condition puts all its mass on $\mathbf{z}_{\mathrm{ML}}$ which minimizes the distortion. However, the network would require infinitely many bits to precisely represent this delta function, and thus the rate goes to infinity.

## 5 RELATED WORKS

**Evaluation of Implicit Generative Models** . Quantitative evaluation of the performance of GANs has been a challenge for the field since the beginning. Many heuristic measures have been proposed, such as the Inception score (Salimans et al., 2016) and the Fréchet Inception Distance (FID) (Heusel et al., 2017). One of the main drawbacks of the IS or FID is that a model that simply memorizes the training dataset would obtain a near-optimal score. Another, drawback of these methods is that they use a pretrained ImageNet classifier weights which makes them sensitive to the weights (Barratt & Sharma, 2018) of the classifier, and less applicable to other domains and datasets. Another evaluation method that sometimes is being used is the Parzen window estimate, which can be shown to be an instance of AIS with zero intermediate distributions, and thus has a very large variance. Another evaluation method of GANs that was proposed in Metz et al. (2018) is measuiring the ability of the generator network to reconstruct the samples from the data distribution. This metric is similar to the distortion obtained at the high-rate regime of our rate distortion framework when $\beta \to \infty$. Another related work is GILBO (Alemi & Fischer, 2018), which similar to our framework does not require the generative model to have a tractable posterior and thus allows direct comparison of VAEs and GANs. However, GILBO can only evaluate the performance of the generative model on the simulated data and not the true data distribution.

**Rate Distortion Theory and Generative Models.** Perhaps the closest work to ours is "Fixing a Broken ELBO" (Alemi et al., 2018), which plots rate-prior distortion curves for VAEs. Our work is different than Alemi et al. (2018) in two key aspects. First, in Alemi et al. (2018) the rate-prior distortion function is evaluated by fixing the architecture of the neural network, and learning the distortion measure $d(\mathbf{x}, f(\mathbf{z}))$ in addition to learning $q(\mathbf{z}|\mathbf{x})$. Whereas, in our definition of rate distortion, we assumed the distortion measure is fixed and given by a trained generative model. As the result, we plot the rate-prior distortion curve for a particular generative model, rather than a particular architecture. The second key difference is that, consistent with the Shannon's rate distortion

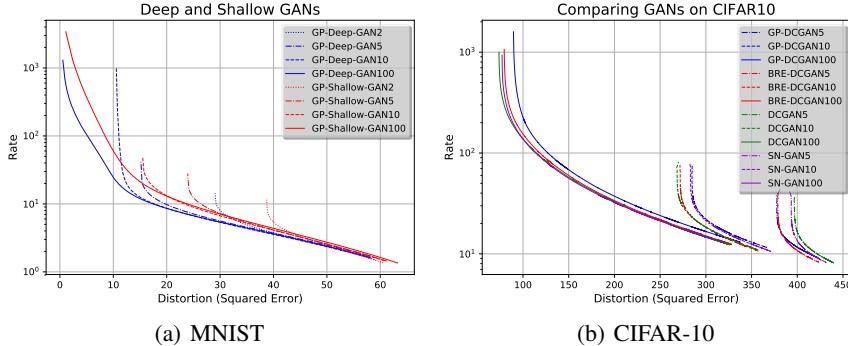

(a) MNIST        (b) CIFAR-10

Figure 3: The rate distortion curves of GANs.

theorem, we find the optimal channel conditional $q^*(\mathbf{z}|\mathbf{x})$ by using AIS; while in Alemi et al. (2018), $q(\mathbf{z}|\mathbf{x})$ is a variational distribution that is restricted to a variational family.

See Appendix A for a discussion of related works about practical compression schemes, distortion-perception tradeoffs, and precision-recall tradeoffs.

## 6 EXPERIMENTS

In this section, we use our rate distortion approximations to answer the following questions: How do different generative models such as VAEs, GANs and AAEs perform at different lossy compression rates? What insights can we obtain from the rate distortion curves about different characteristics of generative models? What is the effect of the code size (width), depth of the network, or the learning algorithm on the rate distortion tradeoffs?

### 6.1 RATE DISTORTION CURVES OF DEEP GENERATIEV MODELS

**Rate Distortion Curves of GANs.** Fig. 3 shows rate distortion curves for GANs trained on MNIST and CIFAR-10. We varied the dimension of the noise vector $\mathbf{z}$, as well as the depth of the decoder. For the GAN experiments on MNIST (Fig. 3a), the label "deep" corresponds to three hidden layers of size 1024, and the label "shallow" corresponds to one hidden layer of size 1024. We trained shallow and deep GANs with Gradient Penalty (GAN-GP) (Gulrajani et al., 2017) with the code size $d \in \{2, 5, 10, 100\}$ on MNIST. For the GAN experiments on CIFAR-10 (Fig. 3b), we trained the DCGAN (Radford et al., 2015), GAN with Gradient Penalty (GP) (Gulrajani et al., 2017), SN-GAN (Miyato et al., 2018), and BRE-GAN (Cao et al., 2018), with the code size of $d \in \{2, 10, 100\}$. In both the MNIST and CIFAR experiments, we observe that in general increasing the code size has the effect of *extending the curve leftwards*. This is expected, since the high-rate regime is effectively measuring reconstruction ability, and additional dimensions in $\mathbf{z}$ improves the reconstruction.

We can also observe from Fig. 3a that increasing the depth pushes the curves *down and to the left*. In other words, the distortion in both high-rate and mid-rate regimes improves. In these regimes, increasing the depth increases the capacity of the network, which enables the network to make a better use of the information in the code space. In the low-rate regime, however, increasing the depth, similar to increasing the latent size, does not improve the distortion.

**Rate Distortion Curves of VAEs.** Fig. 4 compares VAEs, AAEs and GP-GANs (Gulrajani et al., 2017) with the code size of $d \in \{2, 10, 100\}$, and the same decoder architecture on the MNIST dataset. In general, we can see that in the mid-rate to high-rate regimes, VAEs achieve better distortions than GANs with the same architecture. This is expected as the VAE is trained with the ELBO objective, which encourages good reconstructions (in the case of factorized Gaussian decoder). We can see from Fig. 4 that in VAEs, increasing the latent capacity pushes the

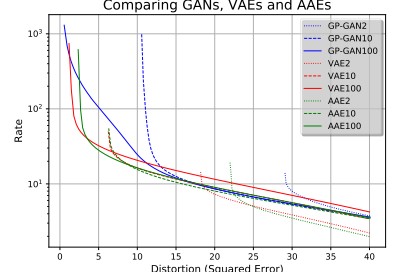

Figure 4: RD curves of VAEs, GANs, AAEs.

rate distortion curve *up and to the left*. In other words, in contrast with GANs where increasing the latent capacity always improves the rate distortion curve, in VAEs, there is a trade-off whereby

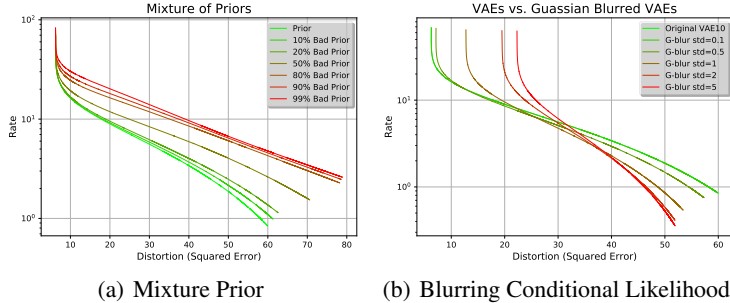

(a) Mixture Prior        (b) Blurring Conditional Likelihood

Figure 5: (a) Effect of damaging the prior of VAE by using a mixture with a bad prior. (b) Effect damaging the conditional likelihood of VAE by adding a Gaussian kernel after the last decoder layer.

increasing the capacity reduces the distortion at the high-rate regime, at the expense of increasing the distortion in the low-rate regime (or equivalently, increasing the rate required to adequately approximate the data).

We believe the performance drop of VAEs in the low-rate regime is symptomatic of the "holes problem" (Rezende & Viola, 2018; Makhzani et al., 2015) in the code space of VAEs with large code size: because these VAEs allocate a large fraction of their latent spaces to garbage images, it requires many bits to get close to the image manifold. Interestingly, this trade-off could also help explain the well-known problem of blurry samples from VAEs: in order to avoid garbage samples (corresponding to large distortion in the low-rate regime), one needs to reduce the capacity, thereby increasing the distortion at the high-rate regime. By contrast, GANs do not suffer from this tradeoff, and one can train high-capacity GANs without sacrificing performance in the low-rate regime.

**Rate Distortion Curves of AAEs.** The AAE was introduced by Makhzani et al. (2015) to address the holes problem of VAEs, by directly matching the aggregated posterior to the prior in addition to optimizing the reconstruction cost. Fig. 4 shows the RD curves of AAEs. In comparison to GANs, AAEs can match the low-rate performane of GANs, but achieve a better high-rate performance. This is expected as AAEs directly optimize the reconstruction cost as part of their objective. In comparison to VAEs, AAEs perform slightly worse at the high-rate regime, which is expected as the adversarial regularization of AAEs is stronger than the KL regularization of VAEs. But AAEs perform slightly better in the low-rate regime, as they can alleviate the holes problem to some extent.

## 6.2 DISTINGUISHING DIFFERENT FAILURE MODES IN GENERATIVE MODELING

Since log-likelihoods constitute only a scalar value, they are unable to distinguish different aspects of a generative model which could be good or bad, such as the prior or the observation model. Here, we show that two manipulations which damage a trained VAE in different ways result in very different behavior of the RD curves.

Our first manipulation, originally proposed by Theis et al. (2015), is to use a mixture of the VAE's density and another distribution concentrated away from the data distribution. As pointed out by Theis et al. (2015), this results in a model which achieves high log-likelihood while generating poor samples. Specifically, after training the VAE10 on MNIST, we "damage" its prior $p(\mathbf{z}) = \mathcal{N}(0, \mathbf{I})$ by altering it to a mixture prior $(1 - \alpha)p(\mathbf{z}) + \alpha q(\mathbf{z})$, where $q(\mathbf{z}) = \mathcal{N}(0, 10\mathbf{I})$ is a "poor" prior, which is chosen to be far away from the original prior $p(\mathbf{z})$; and $\alpha$ is close to 1. This process would results in a "poor" generative model that generates garbage samples most of the time (more precisely with the probability of $\alpha$). Suppose $p(\mathbf{x})$ and $q(\mathbf{x})$ are the likelihood of the good and the poor generative models. It is straightforward to see that $\log q(\mathbf{x})$ is at most $4.6$ nats worse that $\log p(\mathbf{x})$, and thus log-likelihood fails to tell these models apart:

$$\log q(\mathbf{x}) = \log \left( 0.01 p(\mathbf{x}) + 0.99 \int q(\mathbf{z}) p(\mathbf{x}|\mathbf{z}) d\mathbf{z} \right) > \log(0.01 p(\mathbf{x})) \approx \log p(\mathbf{x}) - 4.6 \quad (18)$$

Fig. 5a plots the rate distortion curves of this model for different values of $\alpha$. We can see that the high-rate and log-likelihood performance of the good and poor generative models are almost identical, whereas in the low-rate regime, the RD curves show significant drop in the performance and successfully detect this failure mode of log-likelihood.

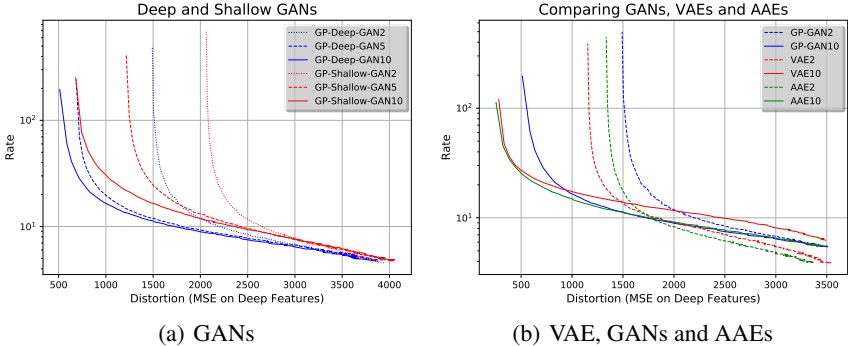

(a) GANs             (b) VAE, GANs and AAEs

Figure 6: The RD curves of GANs, VAEs and AAEs with MSE distortion on the deep feature space. The behavior is qualitatively similar to the results for MSE in images (see Fig. 3 and Fig. 4), suggesting that the RD analysis is not particularly sensitive to the particular choice of metric.

Our second manipulation is to damage the decoder by adding a Gaussian blur kernel after the output layer. Fig. 5b shows the rate distortion curves for different standard deviations of the Gaussian kernel. We can see that, in contrast to the mixture prior experiment, the high-rate performance of the VAE drops due to inability of the decoder to output sharp images. However, we can also see an improvement in the low-rate performance of the VAE. This is because (similarly to log-likelihoods with Gaussian observation models) the data distribution does not necessarily achieve the minimal distortion, and in fact, in the extremely low-rate regime, blurring appears to help by reducing the average Euclidean distance between low-rate reconstructions and the input images. Hence, our two manipulations result in very different effects to the RD curves, suggesting that these curves provide a much richer picture of the performance of generative models, compared to scalar log-likelihoods.

## 6.3 Beyond Pixelwise Mean Squared Error

The experiments discussed above all used pixelwise MSE as the distortion metric. However, for natural images, one could use more perceptually valid distortion metrics such as SSIM (Wang et al., 2004), MSSIM (Wang et al., 2003), or distances between deep features of a CNN (Johnson et al., 2016). Fig. 6 shows the RD curves of GANs, VAEs, and AAEs on the MNIST dataset, using the MSE on the deep features of a CNN as distortion metric. In all cases, the qualitative behavior of the RD curves with this distortion metric closely matches the qualitative behaviors for pixelwise MSE. We can see from Fig. 6a that similar to the RD curves with MSE distortion, GANs with different depths and code sizes have the same low-rate performance, but as the model gets deeper and wider, the RD curves are pushed down and extended to the left. Similarly, we can see from Fig. 6b that compared to GANs and AAEs, VAEs generally have a better high-rate performance, but worse low-rate performance. The fact that the qualitative behaviors of RD curves with this metric closely match those of pixelwise MSE indicates that the results of our analysis are not overly sensitive to the particular choice of distortion metric.

## 7 Conclusion

In this work, we studied rate distortion approximations for evaluating different generative models such as VAEs, GANs and AAEs. We showed that rate distortion curves provide more insights about the model than the log-likelihood alone while requiring roughly the same computational cost. For instance, we observed that while VAEs with larger code size can generally achieve better lossless compression rates, their performances drop at lossy compression in the low-rate regime. Conversely, expanding the capacity of GANs appears to bring substantial reductions in distortion at the high-rate regime without any corresponding deterioration in quality in the low-rate regime. This may help explain the success of large GAN architectures (Brock et al., 2019; Karras et al., 2018a;b). We also discovered that increasing the capacity of GANs by increasing the code size (width) has a very different effect than increasing the depth. The former extends the rate distortion curves leftwards, while the latter pushes the curves down. We also found that different GAN variants with the same code size has almost similar rate distortion curves, and that the code size dominates the algorithmic differences of GANs. Overall, lossy compression yields a richer and more complete picture of the distribution modeling performance of generative models. The ability to quantitatively measure performance tradeoffs should lead to algorithmic insights which can improve these models.

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

## APPENDIX A  RELATED WORKS

**Practical Compression Schemes.**  We have justified our use of compression terminology in terms of Shannon's fundamental result implying that there exist a rate distortion code for any rate distortion pair that is achievable according to the rate distortion function. Interestingly, for *lossless* compression with generative models, there is a practical compression scheme which nearly achieves the theoretical rate (i.e. the negative ELBO): bits-back encoding. The basic scheme was proposed by Wallace (1990); Hinton & Van Camp (1993), and later implemented by Frey & Hinton (1996). Practical versions for modern deep generative models were developed by Townsend et al. (2019); Kingma et al. (2019). We do not currently know of an analogous practical scheme for lossy compression with deep generative models. Other researchers have developed practical coding schemes achieving variational rate distortion bounds for particular latent variable models which exploited the factorial structure of the variational posterior (Ballé et al., 2018; Theis et al., 2017). These methods are not directly applicable in our setting, since we don't assume an explicit encoder network, and our variational posteriors lack a convenient factorized form. We don't know whether our variational approximation will lead to a practical lossy compression scheme, but the successes for other variational methods give us hope.

**Relationship with the Rate-Distortion-Perception Tradeoff.**  Our work is related to Blau & Michaeli (2019) which incorporates a perceptual quality loss function in the rate-distortion framework and characterizes the triple tradeoff between rate distortion and perception. More specifically, Blau & Michaeli (2019) defines the perceptual loss using a divergence between the marginal reconstruction distribution and the data distribution. This perceptual loss is then incorporated as an additional constraint in the rate-distortion framework to encourage the reconstruction distribution to perceptually look like the data distribution. It is shown that as the perceptual constraint becomes tighter, the rate-distortion function elevates more. In our rate-prior distortion framework, we are also enforcing a perceptual constraint on the reconstruction distribution by incorporating the regularization term of $\mathrm{KL}(q(\mathbf{z})\|p(\mathbf{z}))$ in the rate-distortion objective, which encourages matching the aggregated posterior to the prior (Makhzani et al., 2015). More precisely, let us define the reconstruction distribution $r(\mathbf{x})$ as the the distribution obtained by passing the data distribution through the encoder and then the decoder:

$$r(\mathbf{x}, \mathbf{z}) = q(\mathbf{z})p(\mathbf{x}|\mathbf{z}), \qquad \hat{\mathbf{x}} \sim r(\mathbf{x}) = \int_{\mathbf{z}} r(\mathbf{x}, \mathbf{z})d\mathbf{z}. \tag{19}$$

It can be shown that the regularization term $\mathrm{KL}(q(\mathbf{z})\|p(\mathbf{z}))$ upper bounds $\mathrm{KL}(r(\mathbf{x})\|p(\mathbf{x}))$:

$$\mathrm{KL}(q(\mathbf{z})\|p(\mathbf{z})) = \mathrm{KL}(r(\mathbf{x}, \mathbf{z})\|p(\mathbf{x}, \mathbf{z})) \geq \mathrm{KL}(r(\mathbf{x})\|p(\mathbf{x})). \tag{20}$$

In other words, in the rate-prior distortion optimization, for a given distortion constraint, we are not only interested in minimizing the rate $\mathcal{I}(\mathbf{x}; \mathbf{z})$, but also at the same time, we are interested in preserving the perceptual quality of the reconstruction distribution by matching it to the model distribution. In the low-rate regime, when the model is allowed to have large distortions, the model obtains small rates and at the same time preserves the perceptual distribution of the reconstruction samples. As the distortion constraint becomes tighter, the model starts to trade off the rate $\mathcal{I}(\mathbf{x}; \mathbf{z})$ and the perceptual quality $\mathrm{KL}(q(\mathbf{z})\|p(\mathbf{z}))$, which results in an elevated rate distortion curve.

**Relationship with the Precision-Recall Tradeoff.**  One of the main drawbacks of the IS or FID is that they can only provide a single scalar value that cannot distinguish the mode dropping behavior from the mode inventing behavior (generating outlier or garbage samples) in generative models. In order to address this issue, Lucic et al. (2018); Sajjadi et al. (2018) propose to study the precision-recall tradoff for evaluating generative models. In this context, high *precision* implies that the samples from the model distribution are close to the data distribution, and high *recall* implies the generative model can reconstruct (or generate a sample similar to) any sample from the data distribution. The precision-recall curves enable us to identify both the mode dropping and the mode inventing behavior of the generative model. More specifically, mode dropping drops the precision of the model at the high-recall regime, and mode inventing drops the precision of the model in the low-recall regime. Our rate-prior distortion framework can be thought as the information theoretic analogue of the precision-recall curves, which extends the scalar notion of log-likelihood to rate distortion curves. More specifically, in our framework, mode dropping drops the distortion performance of the model at the high-rate regime, and mode inventing drops the distortion performance of the model at the low-rate regime. In Section 6, we will empirically study the effect of mode dropping and mode inventing on our rate-prior distortion curves.

## APPENDIX B    AIS VALIDATION EXPERIMENTS

**AIS Settings.** All the AIS settings including the HMC parameters are provided in Appendix D.2.

**AIS Validation.** We conducted several experiments to validate the correctness of our implementation and the accuracy of the AIS estimates. Firstly, we compared our AIS results with the analytical solution of rate-prior distortion curve on a linear VAE (derived in Appendix D.3.1) trained on MNIST. As shown in Fig. 7, the RD curve estimated by AIS agrees closely with the analytical solution. Secondly, for the main experiments of the paper, we evaluated the tightness of the AIS estimate by computing the BDMC gap. The largest BDMC gap for VAEs and AAEs was 0.127 nats, and the largest BDMC gap for GANs was 1.649 nats, showing that our AIS upper bounds are tight. More details are provided in Appendix D.3.2.

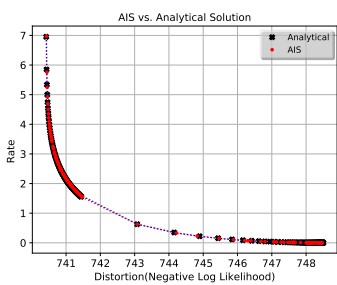

Figure 7: Analytical vs. AIS rate distortion curves for a linear VAE.

## APPENDIX C    PROOFS

### C.1    PROOF OF PROP. 1.

**Proof of Prop. 1a.** As $D$ increases, $\mathcal{R}_p(D)$ is minimized over a larger set, so $\mathcal{R}_p(D)$ is non-increasing function of $D$.

The distortion $\mathbb{E}_{q(\mathbf{x},\mathbf{z})}[d(\mathbf{x}, f(\mathbf{z}))]$ is a linear function of the channel conditional distribution $q(\mathbf{z}|\mathbf{x})$. The mutual information is a convex function of $q(\mathbf{z}|\mathbf{x})$. The $\mathrm{KL}(q(\mathbf{z})\|p(\mathbf{z}))$ is also convex function of $q(\mathbf{z})$, which itself is a linear function of $q(\mathbf{z}|\mathbf{x})$. Thus the rate-prior objective is a convex function of $q(\mathbf{z}|\mathbf{x})$. Suppose for the distortions $D_1$ and $D_2$, $q_1(\mathbf{z}|\mathbf{x})$ and $q_2(\mathbf{z}|\mathbf{x})$ achieve the optimal rates in Eq. 5 respectively. Suppose the conditional $q_\lambda(\mathbf{z}|\mathbf{x})$ is defined as $q_\lambda(\mathbf{z}|\mathbf{x}) = \lambda q_1(\mathbf{z}|\mathbf{x}) + (1 - \lambda)q_2(\mathbf{z}|\mathbf{x})$. The rate-prior objective that the conditional $q_\lambda(\mathbf{z}|\mathbf{x})$ achieves is $\mathcal{I}_\lambda(\mathbf{z};\mathbf{x}) + \mathrm{KL}(q_\lambda(\mathbf{z})\|p(\mathbf{z}))$, and the distortion $D_\lambda$ that this conditional achieves is $D_\lambda = \lambda D_1 + (1 - \lambda)D_2$. Now we have

$$\mathcal{R}_p(D_\lambda) \leq \mathcal{I}_\lambda(\mathbf{z};\mathbf{x}) + \mathrm{KL}(q_\lambda(\mathbf{z})\|p(\mathbf{z})) \tag{21}$$
$$\leq \lambda\mathcal{I}_1(\mathbf{z};\mathbf{x}) + \lambda\mathrm{KL}(q_1(\mathbf{z})\|p(\mathbf{z})) + (1 - \lambda)\mathcal{I}_2(\mathbf{z};\mathbf{x}) + (1 - \lambda)\mathrm{KL}(q_2(\mathbf{z})\|p(\mathbf{z}))$$
$$= \lambda\mathcal{R}_p(D_1) + (1 - \lambda)\mathcal{R}_p(D_2)$$

which proves the convexity of $\mathcal{R}_p(D)$.

**Alternative Proof of Prop. 1a.** We know the rate-prior term $\mathbb{E}_{p_d(\mathbf{x})}\mathrm{KL}(q(\mathbf{z}|\mathbf{x})\|p(\mathbf{z}))$ is a convex function of $q(\mathbf{z}|\mathbf{x})$, and $\mathbb{E}_{q(\mathbf{x},\mathbf{z})}[d(\mathbf{x}, f(\mathbf{z}))]$ is a linear and thus convex function of $q(\mathbf{z}|\mathbf{x})$. As the result, the following optimization problem is a convex optimization problem.

$$\min_{q(\mathbf{z}|\mathbf{x})} \mathbb{E}_{p_d(\mathbf{x})}\mathrm{KL}(q(\mathbf{z}|\mathbf{x})\|p(\mathbf{z})) \qquad s.t. \ \mathbb{E}_{q(\mathbf{x},\mathbf{z})}[d(\mathbf{x}, f(\mathbf{z}))] \leq 0. \tag{22}$$

The rate distortion function $\mathcal{R}_p(D)$ is the perturbation function of the convex optimization problem of Eq. 22. The convexity of $\mathcal{R}_p(D)$ follows from the fact that the perturbation function of any convex optimization problem is a convex function (Boyd & Vandenberghe, 2004).

**Proof of Prop. 1b.** We have

$$\min_{p(\mathbf{z})} \mathcal{R}_p(D) = \min_{p(\mathbf{z})} \min_{q(\mathbf{z}|\mathbf{x}):\mathbb{E}[d(\mathbf{x},f(\mathbf{z}))]\leq D} \mathcal{I}(\mathbf{x};\mathbf{z}) + \mathrm{KL}(q(\mathbf{z})\|p(\mathbf{z})) \tag{23}$$

$$= \min_{q(\mathbf{z}|\mathbf{x}):\mathbb{E}[d(\mathbf{x},f(\mathbf{z}))]\leq D} \min_{p(\mathbf{z})} \mathcal{I}(\mathbf{x};\mathbf{z}) + \mathrm{KL}(q(\mathbf{z})\|p(\mathbf{z})) \tag{24}$$

$$= \min_{q(\mathbf{z}|\mathbf{x}):\mathbb{E}[d(\mathbf{x},f(\mathbf{z}))]\leq D} \mathcal{I}(\mathbf{x};\mathbf{z}) \tag{25}$$

$$= \mathcal{R}(D). \tag{26}$$

where in Eq. 24, we have used the fact that for any function $f(x, y)$, we have

$$\min_x \min_y f(x, y) = \min_y \min_x f(x, y) = \min_{x,y} f(x, y), \tag{27}$$

and in Eq. 25, we have used the fact that $\mathrm{KL}(q(\mathbf{z})\|p(\mathbf{z}))$ is minimized when $p(\mathbf{z}) = q(\mathbf{z})$.

**Proof of Prop. 1c.** In Prop. 1a, we showed that the rate-prior term is a convex function of $q(\mathbf{z}|\mathbf{x})$, and that the distortion is a linear function of $q(\mathbf{z}|\mathbf{x})$. So the summation of them in Eq. 9 will be a convex function of $q(\mathbf{z}|\mathbf{x})$. The unique global optimum of this convex optimization can be found by rewriting Eq. 9 as

$$\mathrm{KL}(q(\mathbf{z}|\mathbf{x})\|p(\mathbf{z})) + \beta\mathbb{E}_{q(\mathbf{z}|\mathbf{x})}[d(\mathbf{x}, f(\mathbf{z}))] = \mathrm{KL}(q(\mathbf{z}|\mathbf{x})\|\frac{1}{Z(\mathbf{x})}p(\mathbf{z})\exp(-\beta d(\mathbf{x}, f(\mathbf{z})))) - \log Z_\beta(\mathbf{x})$$
(28)

where $Z_\beta(\mathbf{x}) = \int p(\mathbf{z})\exp(-\beta d(\mathbf{x}, f(\mathbf{z})))d\mathbf{z}$. The minimum of Eq. 28 is obtained when the KL divergence is zero. Thus the optimal channel conditional is $q_\beta^*(\mathbf{z}|\mathbf{x}) = \frac{1}{Z_\beta(\mathbf{x})}p(\mathbf{z})\exp(-\beta d(\mathbf{x}, f(\mathbf{z})))$.

## C.2 Proof of Prop. 2.

**Proof of Prop. 2a.** $\mathcal{R}(D) \leq \mathcal{R}_p(D)$ was proved in Prop. 1b. To prove the first inequality, note that the summation of rate and distortion is

$$\mathcal{R}_p(D) + D = \mathcal{I}(\mathbf{z}; \mathbf{x}) + \mathbb{E}_{q^*(\mathbf{x},\mathbf{z})}[-\log p(\mathbf{x}|\mathbf{z})] = \mathcal{H}_d + \mathbb{E}_{q^*(\mathbf{z})}\mathrm{KL}(q^*(\mathbf{x}|\mathbf{z})\|p(\mathbf{x}|\mathbf{z})) \geq \mathcal{H}_d.$$
(29)

where $q^*(\mathbf{x}, \mathbf{z})$ is the optimal joint channel conditional, and $q^*(\mathbf{z})$ and $q^*(\mathbf{x}|\mathbf{z})$ are its marginal and conditional. The equality happens if there is a joint distribution $q(\mathbf{x}, \mathbf{z})$, whose conditional $q(\mathbf{x}|\mathbf{z}) = p(\mathbf{x}|\mathbf{z})$, and whose marginal over $\mathbf{x}$ is $p_d(\mathbf{x})$. But note that such a joint distribution might not exist for an arbitrary $p(\mathbf{x}|\mathbf{z})$.

**Proof of Prop. 2b.** The proof can be easily obtained by using $d(\mathbf{x}, f(\mathbf{z})) = -\log p(\mathbf{x}|\mathbf{z})$ in Prop. 1c.

**Proof of Prop. 2c.** Based on Prop. 2b, at $\beta = 1$, we have

$$Z_\beta^*(\mathbf{x}) = \int p(\mathbf{z})p(\mathbf{x}|\mathbf{z})d\mathbf{z} = p(\mathbf{x}).$$
(30)

## C.3 Proof of Prop. 3.

The set of pairs of $\left(R_k^{\mathrm{AIS}}(\mathbf{x}), D_k^{\mathrm{AIS}}(\mathbf{x})\right)$ are achievable rate-prior distortion pairs (achieved by $q_k^{\mathrm{AIS}}(\mathbf{z}|\mathbf{x})$). Thus, by the definition of $\mathcal{R}_p(D)$, $\mathcal{R}_p^{\mathrm{AIS}}(D)$ falls in the achievable region of $\mathcal{R}_p(D)$ and, thus maintains an upper bound on it: $\mathcal{R}_p^{\mathrm{AIS}}(D) \geq \mathcal{R}_p(D)$.

## C.4 Proof of Prop. 4.

AIS has the property that for any step $k$ of the algorithm, the set of chains up to step $k$, and the partial computation of their weights, can be viewed as the result of a complete run of AIS with target distribution $q_k^*(\mathbf{z}|\mathbf{x})$. Hence, we assume without loss of generality that we are looking at a complete run of AIS (but our analysis applies to the intermediate distributions as well).

Let $q_k^{\mathrm{AIS}}(\mathbf{z}|\mathbf{x})$ denote the distribution of final samples produced by AIS. More precisely, it is a distribution encoded by the following procedure:

1. For each data point $\mathbf{x}$, we run $M$ independent AIS chains, numbered $i = 1, \ldots, M$. Let $\mathbf{z'}_k^i$ denotes the $k$-th state of the $i$-th chain. The joint distribution of the forward pass up to the $k$-th state is denoted by $q_f(\mathbf{z'}_1^i, \ldots, \mathbf{z'}_k^i|\mathbf{x})$. The un-normalized joint distribution of the backward pass is denoted by

$$\tilde{q}_b(\mathbf{z'}_1^i, \ldots, \mathbf{z'}_k^i|\mathbf{x}) = p(\mathbf{z'}_k^i)\exp(-\beta_k d(\mathbf{x}, f(\mathbf{z'}_k^i)))\, q_b(\mathbf{z'}_1^i, \ldots, \mathbf{z'}_{k-1}^i|\mathbf{z'}_k^i, \mathbf{x}).$$
(31)

2. Compute the importance weights and normalized importance weights of each chain using $w_k^i = \frac{\tilde{q}_b(\mathbf{z'}_1^i, \ldots, \mathbf{z'}_k^i|\mathbf{x})}{q_f(\mathbf{z'}_1^i, \ldots, \mathbf{z'}_k^i|\mathbf{x})}$ and $\tilde{w}_k^i = \frac{w_k^i}{\sum_{i=1}^M w_k^i}$.

3. Select a chain index $S$ with probability of $\tilde{w}_k^i$.

4. Assign the selected chain values to $(\mathbf{z}_1^1, \ldots, \mathbf{z}_k^1)$:

$$(\mathbf{z}_1^1, \ldots, \mathbf{z}_k^1) = (\mathbf{z'}_1^S, \ldots, \mathbf{z'}_k^S). \tag{32}$$

5. Keep the unselected chain values and re-label them as $(\mathbf{z}_1^{2:M}, \ldots, \mathbf{z}_k^{2:M})$:

$$(\mathbf{z}_1^{2:M}, \ldots, \mathbf{z}_k^{2:M}) = (\mathbf{z'}_1^{-S}, \ldots, \mathbf{z'}_k^{-S}). \tag{33}$$

where $-S$ denotes the set of all indices except the selected index $S$.

6. Return $\mathbf{z} = \mathbf{z}_k^1$.

More formally, the AIS distribution is

$$q_k^{\text{AIS}}(\mathbf{z}|\mathbf{x}) = \mathbb{E}_{\prod_{i=1}^M q_f(\mathbf{z'}_1^i, \ldots, \mathbf{z'}_k^i | \mathbf{x})} \Big[ \sum_{i=1}^M \tilde{w}_k^i \delta(\mathbf{z} - \mathbf{z'}_k^i) \Big]. \tag{34}$$

Using the AIS distribution $q_k^{\text{AIS}}(\mathbf{z}|\mathbf{x})$ defined as above, we define the AIS distortion $D_k^{\text{AIS}}(\mathbf{x})$ and the AIS rate-prior $R_k^{\text{AIS}}(\mathbf{x}) = \text{KL}(q_k^{\text{AIS}}(\mathbf{z}|\mathbf{x}) \| p(\mathbf{z}))$ as follows:

$$D_k^{\text{AIS}}(\mathbf{x}) = \mathbb{E}_{q_k^{\text{AIS}}(\mathbf{z}|\mathbf{x})}[d(\mathbf{x}, f(\mathbf{z}))] \tag{35}$$

$$R_k^{\text{AIS}}(\mathbf{x}) = \text{KL}(q_k^{\text{AIS}}(\mathbf{z}|\mathbf{x}) \| p(\mathbf{z})). \tag{36}$$

In order to estimate $R_k^{\text{AIS}}(\mathbf{x})$ and $D_k^{\text{AIS}}(\mathbf{x})$, we define

$$\hat{D}_k^{\text{AIS}}(\mathbf{x}) = \sum_{i=1}^M \tilde{w}_k^i d(\mathbf{x}, f(\mathbf{z'}_k^i)), \tag{37}$$

$$\hat{Z}_k^{\text{AIS}}(\mathbf{x}) = \frac{1}{M} \sum_{i=1}^M w_k^i, \tag{38}$$

$$\hat{R}_k^{\text{AIS}}(\mathbf{x}) = -\log \hat{Z}_k^{\text{AIS}}(\mathbf{x}) - \beta_k \hat{D}_k^{\text{AIS}}(\mathbf{x}). \tag{39}$$

We would like to prove that

$$\mathbb{E}_{\prod_{i=1}^M q_f(\mathbf{z'}_1^i, \ldots, \mathbf{z'}_k^i | \mathbf{x})}[\hat{D}_k^{\text{AIS}}(\mathbf{x})] = D_k^{\text{AIS}}(\mathbf{x}), \tag{40}$$

$$\mathbb{E}_{\prod_{i=1}^M q_f(\mathbf{z'}_1^i, \ldots, \mathbf{z'}_k^i | \mathbf{x})}[\hat{R}_k^{\text{AIS}}(\mathbf{x})] \geq R_k^{\text{AIS}}(\mathbf{x}). \tag{41}$$

The proof of Eq. 40 is straightforward:

$$
\begin{aligned}
D_k^{\text{AIS}}(\mathbf{x}) &= \mathbb{E}_{q_k^{\text{AIS}}(\mathbf{z}|\mathbf{x})}[d(\mathbf{x}, f(\mathbf{z}))], \\
&= \int q_k^{\text{AIS}}(\mathbf{z}|\mathbf{x}) d(\mathbf{x}, f(\mathbf{z})) d\mathbf{z}, \\
&= \int \mathbb{E}_{\prod_{i=1}^M q_f(\mathbf{z'}_1^i, \ldots, \mathbf{z'}_k^i | \mathbf{x})} \Big[ \sum_{i=1}^M \tilde{w}_k^i \delta(\mathbf{z} - \mathbf{z'}_k^i) \Big] d(\mathbf{x}, f(\mathbf{z})) d\mathbf{z}, \\
&= \mathbb{E}_{\prod_{i=1}^M q_f(\mathbf{z'}_1^i, \ldots, \mathbf{z'}_k^i | \mathbf{x})} \sum_{i=1}^M \tilde{w}_k^i \Big[ \int \delta(\mathbf{z} - \mathbf{z'}_k^i) d(\mathbf{x}, f(\mathbf{z})) d\mathbf{z} \Big], \\
&= \mathbb{E}_{\prod_{i=1}^M q_f(\mathbf{z'}_1^i, \ldots, \mathbf{z'}_k^i | \mathbf{x})} \sum_{i=1}^M \tilde{w}_k^i d(\mathbf{x}, f(\mathbf{z'}_k^i)), \\
&= \mathbb{E}_{\prod_{i=1}^M q_f(\mathbf{z'}_1^i, \ldots, \mathbf{z'}_k^i | \mathbf{x})}[\hat{D}_k^{\text{AIS}}(\mathbf{x})].
\end{aligned}
\tag{42}
$$

Eq. 42 shows that $\hat{D}_k^{\text{AIS}}(\mathbf{x})$ is an unbiased estimate of $D_k^{\text{AIS}}(\mathbf{x})$. We also know $\log \hat{Z}_k^{\text{AIS}}(\mathbf{x})$ obtained by Eq. 38 is the estimate of the log partition function, and by the Jenson's inequality lower bounds in expectation the true log partition function: $\mathbb{E}[\log \hat{Z}_k^{\text{AIS}}(\mathbf{x})] \leq \log Z_k(\mathbf{x})$. After obtaining $\hat{D}_k^{\text{AIS}}(\mathbf{x})$

and $\log \hat{Z}_k^{\text{AIS}}(\mathbf{x})$, we use Eq. 39 to obtain $\hat{R}_k^{\text{AIS}}(\mathbf{x})$. Now, it remains to prove Eq. 41, which states that $\hat{R}_k^{\text{AIS}}(\mathbf{x})$ upper bounds the AIS rate term $R_k^{\text{AIS}}(\mathbf{x})$ in expectation.

Let $q_k^{\text{AIS}}(\mathbf{z}_1^{1:M}, \ldots, \mathbf{z}_k^{1:M}|\mathbf{x})$ denote the joint AIS distribution over all states of $\{\mathbf{z}_1^{1:M}, \ldots, \mathbf{z}_k^{1:M}\}$, defined in Eq. 32 and Eq. 33. It can be shown that (see Domke & Sheldon (2018))

$$q_k^{\text{AIS}}(\mathbf{z}_1^{1:M}, \ldots, \mathbf{z}_k^{1:M}|\mathbf{x}) = \frac{\tilde{q}_b(\mathbf{z}_1^1, \ldots, \mathbf{z}_k^1|\mathbf{x}) \prod_{i=2}^M q_f(\mathbf{z}_1^i, \ldots, \mathbf{z}_k^i|\mathbf{x})}{\hat{Z}_k^{\text{AIS}}(\mathbf{x})} \tag{43}$$

$$= \frac{p(\mathbf{z}_k^1) \exp(-\beta_k d(\mathbf{x}, f(\mathbf{z}_k^1))) \, q_b(\mathbf{z}_1^1, \ldots, \mathbf{z}_{k-1}^1|\mathbf{z}_k^1, \mathbf{x}) \prod_{i=2}^M q_f(\mathbf{z}_1^i, \ldots, \mathbf{z}_k^i|\mathbf{x})}{\hat{Z}_k^{\text{AIS}}(\mathbf{x})}. \tag{44}$$

In order to simplify notation, suppose $\mathbf{z}_k^1$ is denoted by $\mathbf{z}$, and all the other variables $\{\mathbf{z}_1^{1:M}, \ldots, \mathbf{z}_{k-1}^{1:M}, \mathbf{z}_k^{2:M}\}$ are denoted by $\mathbf{V}$. Using this notation, we define $p(\mathbf{V}|\mathbf{z}, \mathbf{x})$ and $q_k^{\text{AIS}}(\mathbf{z}, \mathbf{V}|\mathbf{x})$ as follows:

$$p(\mathbf{V}|\mathbf{z}, \mathbf{x}) := q_b(\mathbf{z}_1^1, \ldots, \mathbf{z}_{k-1}^1|\mathbf{z}_k^1, \mathbf{x}) \prod_{i=2}^M q_f(\mathbf{z}_1^i, \ldots, \mathbf{z}_k^i|\mathbf{x}), \tag{45}$$

$$q_k^{\text{AIS}}(\mathbf{z}, \mathbf{V}|\mathbf{x}) := q_k^{\text{AIS}}(\mathbf{z}_1^{1:M}, \ldots, \mathbf{z}_k^{1:M}|\mathbf{x}) \tag{46}$$

Using the above notation, Eq. 44 can be re-written as

$$\hat{Z}_k^{\text{AIS}}(\mathbf{x}) = \frac{p(\mathbf{z}) \exp(-\beta_k d(\mathbf{x}, f(\mathbf{z}))) \, p(\mathbf{V}|\mathbf{z}, \mathbf{x})}{q_k^{\text{AIS}}(\mathbf{z}, \mathbf{V}|\mathbf{x})}. \tag{47}$$

Hence,

$$\begin{aligned}
\mathbb{E}[\log \hat{Z}_k^{\text{AIS}}(\mathbf{x})] &= \mathbb{E}[\log p(\mathbf{z}) - \log q_k^{\text{AIS}}(\mathbf{z}, \mathbf{V}|\mathbf{x}) + \log p(\mathbf{V}|\mathbf{x}, \mathbf{z})] - \beta_k \mathbb{E}[d(\mathbf{x}, f(\mathbf{z}))] \\
&= -\text{KL}(q_k^{\text{AIS}}(\mathbf{z}, \mathbf{V}|\mathbf{x}) \| p(\mathbf{z}) p(\mathbf{V}|\mathbf{z}, \mathbf{x})) - \beta_k \mathbb{E}[d(\mathbf{x}, f(\mathbf{z}))] \\
&\leq -\text{KL}(q_k^{\text{AIS}}(\mathbf{z}|\mathbf{x}) \| p(\mathbf{z})) - \beta_k \mathbb{E}[d(\mathbf{x}, f(\mathbf{z}))],
\end{aligned} \tag{48}$$

where the inequality follows from the monotonicity of KL divergence. Rearranging terms, we bound the rate:

$$R_k^{\text{AIS}}(\mathbf{x}) = \text{KL}(q_k^{\text{AIS}}(\mathbf{z}|\mathbf{x}) \| p(\mathbf{z})) \leq -\mathbb{E}[\log \hat{Z}_k^{\text{AIS}}(\mathbf{x})] - \beta_k \mathbb{E}[d(\mathbf{x}, f(\mathbf{z}))] = \mathbb{E}[\hat{R}_k^{\text{AIS}}(\mathbf{x})]. \tag{49}$$

Eq. 49 shows that $\hat{R}_k^{\text{AIS}}(\mathbf{x})$ upper bounds the AIS rate-prior $R_k^{\text{AIS}}(\mathbf{x})$ in expectation. We also showed $\hat{D}_k^{\text{AIS}}(\mathbf{x})$ is an unbiased estimate of the AIS distortion $D_k^{\text{AIS}}(\mathbf{x})$. Hence, the estimated AIS rate-prior curve upper bounds the AIS rate-prior distortion curve in expectation: $\mathbb{E}[\hat{\mathcal{R}}_p^{\text{AIS}}(D)] \geq \mathcal{R}_p^{\text{AIS}}(D)$.

## APPENDIX D  EXPERIMENTAL DETAILS

The code for reproducing all the experiments of this paper will be open sourced publicly.

### D.1  DATASETS AND MODELS

We used MNIST (LeCun et al., 1998) and CIFAR-10 (Krizhevsky & Hinton, 2009) datasets in our experiments.

**Real-Valued MNIST.** For the VAE experiments on the real-valued MNIST dataset (Fig. 4a), we used the "VAE-50" architecture described in (Wu et al., 2016), and only changed the code size in our experiments. The decoder variance is a global parameter learned during the training. The network was trained for 1000 epochs with the learning rate of 0.0001 using the Adam optimizer (Kingma & Ba, 2014).

For the GAN experiments on MNIST (Fig. 3a), we used the "GAN-50" architecture described in (Wu et al., 2016). In order to stabilize the training dynamic, we used the gradient penalty (GP) (Salimans et al., 2016). In our deep architectures, we used code sizes of $d \in \{2, 5, 10, 100\}$ and three hidden

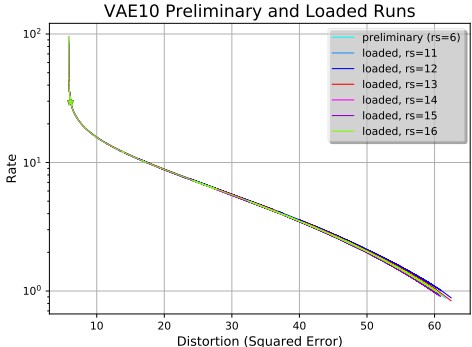

Figure 8: The rate-prior distortion curves obtained by adaptively tuning the HMC parameters in the preliminary run, and pre-loading the HMC parameters in the second formal run. "rs" in the legend indicates the random seed used in the second run.

layers each having 1024 hidden units to obtain the following GAN models: Deep-GAN2, Deep-GAN5, Deep-GAN10 and Deep-GAN100. The shallow GANs architectures are similar to the deep architectures but with one layer of hidden units.

**CIFAR-10.** For the CIFAR-10 experiments (Fig. 3b), we experimented with different GAN models such as DCGAN (Radford et al., 2015), DCGAN with Gradient Penalty (GP-GAN) (Gulrajani et al., 2017), Spectral Normalization (SN-GAN) (Miyato et al., 2018), and DCGAN with Binarized Representation Entropy Regularization (BRE-GAN) (Cao et al., 2018). The numbers at the end of each GAN name in Fig. 3b indicate the code size.

### D.2 AIS SETTINGS FOR RD CURVES

For each RD curve, there are 1999 points computed with only one AIS chain, for 999 $\beta's$ spaced linearly from $\beta_{max}$ to 1 and another 999 $\beta's$ spaced linearly from 1 to $\beta_{min}$, and plus $\beta = 1$, thus 1999 points. $\beta_{min} = \frac{1}{12}$ for all models. $\beta_{max} = \frac{1}{0.0003} \approx 3333$ for 100 dimensional models such as GAN100, VAE100 or AAE 100, and $\beta_{max} = \frac{1}{0.00002770224} \approx 36098$ for the rest (2, 5 and 10 dimensional models). The AIS temperature schedule ($t$ for intermediate distributions) is generated by three steps:

1. Build a sigmoid temperature schedule used in Wu et al. (2016) for $\beta_{max}$ with $N$ intermediate distributions.

2. If there are fewer than 800 intermediate distributions before the first RD point in the temperature schedule initialized in Step 1, overwrite this part of the schedule with 800 intermediate distributions linearly spaced between $\beta_0 = 0$ the first RD point ($\beta_1$).

3. If there are fewer than 10 intermediate distributions between two RD points in the temperature schedule initialized in Step 1, overwrite this part of the schedule with 10 intermediate distributions linearly spaced between two RD points.

For the 2, 5 and 10 dimensional models, $N = 40000$, and the above procedure will result in 60292 intemediate distributions in total. For 100 dimensional models, to ensure accuracy of our AIS estimator with a small BDMC gap, we used $N = 1600000$ and the above procedure will result in 1611463 intermediate distributions in total. We used 20 leap frog steps for HMC, 40 independent chains, on a single batch of 50 images. On MNIST, we also tested with a larger batch of 500 MNIST images, but did not observe significant difference compared with a batch 50 images, thus we did all of our experiments with a single batch 50 images. On a P100 GPU, for MNIST, it takes 4-7 hours to compute an RD curve for 60292 intermediate distributions and takes around 7 days for 1611463 intermediate distributions. For all of the CIFAR experiments, we used the schedule with 60292 intermediate distributions, and each experiment takes about 7 days to complete.

**Adaptive Tuning of HMC Parameters.** While running the AIS chain, the parameters of the HMC kernel cannot be adaptively tuned, since it would violate the Markovian property of the chain. So in

order to be able to adaptively tune HMC parameters such as the number of leapfrog steps and the step size, in all our experiments, we first do a preliminary run where the HMC parameters are adaptively tuned to yield an average acceptance probability of $65\%$ as suggested in Neal (2001). Then in the second "formal" run, we pre-load and fix the HMC parameters found in the preliminary run, and start the chain with a new random seed to obtain our final results. Interestingly, we observed that the difference in the RD curves obtained from the preliminary run and the formal runs with various different random seeds is very small, as shown in Fig. 8. This figure shows that the AIS with the HMC kernel is robust against different choices of random seeds for approximating the RD curve of VAE10.

### D.3 VALIDATION OF AIS EXPERIMENTS

We conducted several experiments to validate the correctness of our implementation and the accuracy of the AIS estimates.

#### D.3.1 ANALYTICAL SOLUTION OF THE RATE-PRIOR DISTORTION OPTIMIZATION ON THE LINEAR VAE

We compared our AIS results with the analytical solution of the rate-prior distortion optimization on a linear VAE trained on MNIST as shown in Fig. 7.

In order to derive the analytical solution, we first find the optimal distribution $q_\beta^*(\mathbf{z}|\mathbf{x})$ from Prop. 2b. For simplicity, we assume a fixed identity covariance matrix $I$ at the output of the conditional likelihood of the linear VAE decoder. In other words, the decoder of the VAE is simply: $\mathbf{x} = \mathbf{Wz} + \mathbf{b} + \epsilon$, where $\mathbf{x}$ is the observation, $\mathbf{z}$ is the latent code vector, $\mathbf{W}$ is the decoder weight matrix and $\mathbf{b}$ is the bias. The observation noise of the decoder is $\epsilon \sim \mathcal{N}(\mathbf{0}, \mathbf{I})$. It's easy to show that the conditional likelihood raised to a power $\beta$ is: $p(\mathbf{x}|\mathbf{z})^\beta = \mathcal{N}(\mathbf{x}|\mathbf{Wz} + \mathbf{b}, \frac{1}{\beta}\mathbf{I})$. Then, $q_\beta^*(\mathbf{z}|\mathbf{x}) = \mathcal{N}(\mathbf{z}|\mu_\beta, \mathbf{\Sigma}_\beta)$, where

$$\mu_\beta = \mathbb{E}_{q_\beta^*(\mathbf{z}|\mathbf{x})}[\mathbf{z}] = \mathbf{W}^\intercal(\mathbf{WW}^\intercal + \beta^{-1}\mathbf{I})^{-1}(\mathbf{x} - \mathbf{b})$$
$$\mathbf{\Sigma}_\beta = \text{Cov}_{q_\beta^*(\mathbf{z}|\mathbf{x})}[\mathbf{z}] = \mathbf{I} - \mathbf{W}^\intercal(\mathbf{WW}^\intercal + \beta^{-1}\mathbf{I})^{-1}\mathbf{W}$$
$$(50)$$

For numerical stability, we can further simplify thw above by taking the SVD of $\mathbf{W}$ : let $\mathbf{W} = \mathbf{UDV}^\intercal$, and then apply the Woodbury Matrix Identity to the matrix inversion, we can get:

$$\boldsymbol{\mu}_\beta = \mathbf{VR}_\beta\mathbf{U}^\intercal(\mathbf{x} - \mathbf{b}) \tag{51}$$
$$\mathbf{\Sigma}_\beta = \mathbf{VS}_\beta\mathbf{V}^\intercal \tag{52}$$

Where $\mathbf{R}_\beta$ is a diagonal matrix with the $i^{th}$ diagonal entry being $\frac{d_i}{d_i^2 + \frac{1}{\beta}}$ and $\mathbf{S}_\beta$ is a diagonal matrix with the $i^{th}$ diagonal entry being $[\frac{1}{\beta d_i^2 + 1}]$, where $d_i$ is the $i^{th}$ diagonal entry of $\mathbf{D}$
Then, the analytical solution for optimum rate is:

$$D_{KL}(q_\beta^*(\mathbf{z}|\mathbf{x})||p(\mathbf{z})) = D_{KL}(\mathcal{N}(\mathbf{z}|\boldsymbol{\mu}_\beta, \mathbf{\Sigma}_\beta)||\mathcal{N}(\mathbf{z}|\mathbf{0}, \mathbf{I})) \tag{53}$$
$$= \frac{1}{2}\left(\text{tr}\left(\mathbf{\Sigma}_\beta\right) + (-\boldsymbol{\mu}_\beta)^\intercal(-\boldsymbol{\mu}_\beta) - k + \ln\left((\det \mathbf{\Sigma}_\beta)^{-1}\right)\right) \tag{54}$$
$$= \frac{1}{2}\left(\text{tr}\left(\mathbf{\Sigma}_\beta\right) + (\boldsymbol{\mu}_\beta)^\intercal(\boldsymbol{\mu}_\beta) - k - \ln\left(\det \mathbf{\Sigma}_\beta\right)\right) \tag{55}$$

Where k is the dimension of the latent code $\mathbf{z}$. With negative log-likelihood as the distortion metric, the analytically form of distortion term is:

$$\mathbb{E}_{q_\beta^*(\mathbf{z}|\mathbf{x})}\left[-\log p(\mathbf{x}|\mathbf{z})\right] \tag{56}$$
$$= \int_{-\infty}^{\infty} -\log((2\pi)^{-k/2}\exp\left\{-\frac{1}{2}(\mathbf{x} - (\mathbf{Wz} + \mathbf{b}))^\intercal(\mathbf{x} - (\mathbf{Wz} + \mathbf{b}))\right\})q_\beta^*(\mathbf{z}|\mathbf{x}d\mathbf{z} \tag{57}$$
$$= -(\log((2\pi)^{-k/2}) + \frac{1}{2}\int_{-\infty}^{\infty}\left\{(\mathbf{x} - (\mathbf{Wz} + \mathbf{b}))^\intercal(\mathbf{x} - (\mathbf{Wz} + \mathbf{b}))\right\}q_\beta^*(\mathbf{z}|\mathbf{x}d\mathbf{z}) \tag{58}$$
$$= \frac{k}{2}\log(2\pi) + \frac{1}{2}(\mathbf{x} - \mathbf{b})^\intercal(\mathbf{x} - \mathbf{b}) - (\mathbf{W}\boldsymbol{\mu}_\beta)^\intercal(\mathbf{x} - \mathbf{b}) + \frac{1}{2}\mathbb{E}_{q_\beta^*(\mathbf{z}|\mathbf{x})}\left[(\mathbf{Wz})^\intercal(\mathbf{Wz})\right] \tag{59}$$

where $\mathbb{E}_{q_\beta^*(\mathbf{z}|\mathbf{x})}\left[(\mathbf{Wz})^\intercal(\mathbf{Wz})\right]$ can be obtained by change of variable: Let $\mathbf{y} = \mathbf{Wz}$, then:

$$\mathbb{E}_{q^*(y)}[\mathbf{y}] = \mathbf{W}\boldsymbol{\mu}_\beta = \mathbf{U}(\mathbf{I} - \mathbf{S}_\beta)\mathbf{U}^\intercal(\mathbf{X} - \mathbf{b}) \tag{60}$$

$$\mathrm{Cov}_{q^*(y)}[\mathbf{y}] = \mathbf{W}\boldsymbol{\Sigma}_\beta\mathbf{W}^\intercal = \mathbf{UDR}_\beta\mathbf{DU}^\intercal \tag{61}$$

$$\mathbb{E}_{q_\beta^*(\mathbf{z}|\mathbf{x})}\left[(\mathbf{Wz})^\intercal(\mathbf{Wz})\right] = \mathbb{E}_{q^*(y)}[\mathbf{y}^\intercal\mathbf{y}] = \mathbb{E}_{q^*(y)}[\mathbf{y}]^\intercal\,\mathbb{E}_{q^*(y)}[\mathbf{y}] + \mathrm{tr}(\mathrm{Cov}_{q^*(y)}[\mathbf{y}]) \tag{62}$$

$$\tag{63}$$

### D.3.2 THE BDMC GAP

We evaluated the tightness of the AIS estimate by computing the BDMC gaps using the same AIS settings. Fig. 9, shows the BDMC gaps at diffrent compression rates for the VAE, GAN and AAE experiments on the MNIST dataset. The largest BDMC gap for VAEs and AAEs is 0.127 nats, and the largest BDMC gap for GANs is 1.649 nats, showing that our AIS upper bounds are tight.

### D.4 HIGH-RATE VS. LOW-RATE RECONSTRUCTIONS

In this section, we visualize the high-rate ($\beta \approx 3500$) and low-rate ($\beta = 0$) reconstructions of the MNIST images for VAEs, GANs and AAEs with different hidden code sizes. The qualitative results are shown in Fig. 10 and Fig. 11, which is in consistent with the quantitative results presented in experiment section of the paper.

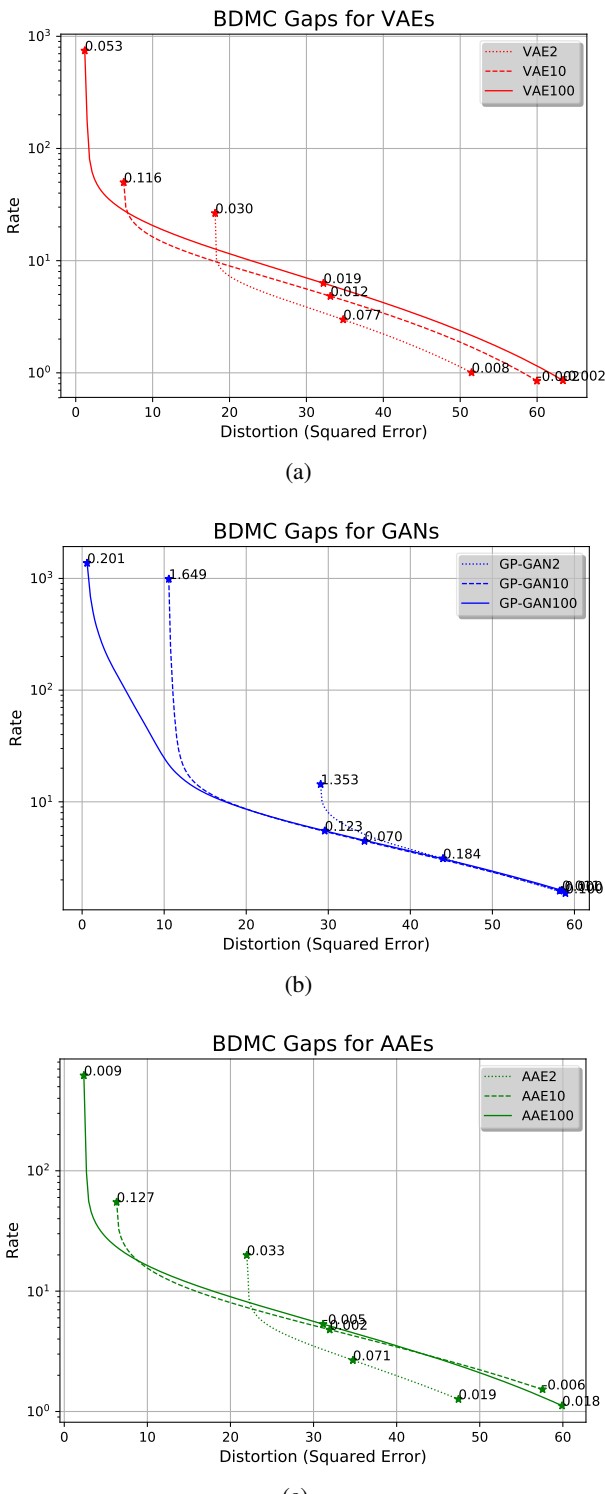

Figure 9: The BDMC gaps annotated on estimated AIS Rate-Prior Distortion curves of (a) VAEs, (b) GANs, and (c) AAEs.

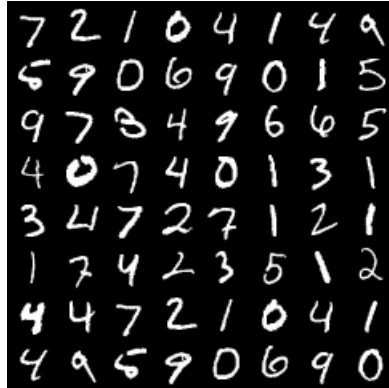

(a) Original MNIST test images

| (b) Low Rate VAE2 | (c) Low Rate AAE2 | (d) Low Rate GAN2 |

| (e) Low Rate VAE10 | (f) Low Rate AAE10 | (g) Low Rate GAN10 |

| (h) Low Rate VAE100 | (i) Low Rate AAE100 | (j) Low Rate GAN100 |

Figure 10: Low-rate reconstructions ($\beta = 0$) of VAEs, GANs and AAEs on MNIST.

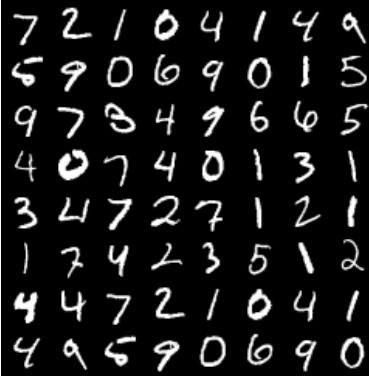

(a) Original MNIST test images.

(b) High Rate VAE2      (c) High Rate AAE2      (d) High Rate GAN2

(e) High Rate VAE10      (f) High Rate AAE10      (g) High Rate GAN10

(h) High Rate VAE100      (i) High Rate AAE100      (j) High Rate GAN100

Figure 11: High-rate reconstructions ($\beta_{\max}$) of VAEs, GANs and AAEs on MNIST. $\beta_{\max} = 3333$ for 100 dimensional models, and $\beta_{\max} = 36098$ for the 2 and 10 dimensional models.

