# OpenReview forum: "Evaluating Lossy Compression Rates of Deep Generative Models"
_ICLR.cc/2020/Conference — Reject_

### Official Review · AnonReviewer2 · 2019-10-18
**Official Blind Review #2**

**Rating:** 8

**Review:**

Summary:

The paper proposes a new way to evaluate generative models that don't have tractable likelihoods, such as VAEs or GANs. Such generative models are composed of a prior over latent variables and a decoder that maps latent variables to data. The idea is to evaluate a trained model in terms of the best (lossy) compression rate that can be achieved by encoding a datapoint (e.g. an image) into the latent space, as a function of a permitted distortion between the datapoint and its reconstruction after decoding. The paper describes a method that estimates an upper bound on this rate-distortion curve using annealed importance sampling. The method is applied in evaluating and comparing a few VAE, GAN and AAE architectures on images (MNIST and CIFAR-10).

Overall evaluation:

This is a very good paper, and I'm happy to recommend it for acceptance.

The problem considered (evaluating generative models with intractable likelihoods) is an interesting and important one. In general, such models are hard to evaluate and compare with each other. The paper proposes a new method for evaluating them, which can also improve our understanding of these models and potentially diagnose them in practice.

The method is well-motivated and backed by theoretical results. One clever aspect of the method is the way annealed importance sampling is used to approximate the rate-distortion curve: instead of sampling separately the rate for each distortion level with a different AIS run, a single AIS run is used to approximate the whole curve. This is done by taking the various points on the curve to correspond to intermediate distributions in AIS, which is quite clever.

The paper is well written, precise, and contains sufficient theoretical background to motivate the method.

The experiments are done carefully, and the results are interesting. I found particularly interesting the fact that VAEs behave differently to GANs (in terms of their rate-distortion tradeoff) when the dimensionality of the latent space is increased.

Some discussion and critical feedback:

I found the paper too long (10 full pages). I appreciate the detail, precision and depth of explanation, but I think it would be good to reduce the amount of text if possible.

I though that the introduction was too specific to VAEs/GANs and to image modelling, which may give the impression that these are the main models/tasks of interest. I understand that these are the models and tasks that the paper is interested in, but I think it would be better if the introduction acknowledged the existence of other types of generative models (e.g. likelihood-based models such as autoregressive models and normalizing flows) and were less specific to image applications.

Proposition 3 is true almost by definition, since R_p is defined to be the minimum of all rate-distortion curves. I wonder if something more informative can be shown here. For example, my understanding is that the reason R^{AIS}_p is not optimal is due to the bias of the importance-sampling estimator in eq. (12). Since this bias asymptotically goes to zero, I suspect that R^{AIS}_p may become equal to R_p for M -> infinity, and perhaps the bound improves monotonically as M increases.

If my understanding is correct, the reason for the inequality in proposition 4 is that log\hat{Z} is a biased estimate of logZ (due to Jensen's inequality) despite \hat{Z} being an unbiased estimate of Z. If that's all there is to it, the proof in the appendix, although precise, is a bit of an overkill. It also seems to me that the log\hat{Z} bias also approaches zero as M -> infinity, so this inequality maybe also becomes an equality asymptotically.

In future work, it would be interesting to also evaluate flow-based models (such as Glow). Since these models give exact likelihoods, it may be good to observe how the evaluation based on rate-distortion curves compares with a likelihood-based evaluation.

Section 6 attributes the performance drop of VAEs in the low-rate regime to the "holes problem". If that's true, then I would expect the situation to be improved with more flexible prior / posterior models. What prior / posterior models were used in the experiments? If only diagonal Gaussians were used, then it would be interesting to see whether more flexible priors / posteriors such as normalizing flows would change the results.

Minor corrections and suggestions for improvement:

"For continuous inputs, the metric is often dominated by the fine-grained distribution over pixels rather than the high-level structure"
This statement is specific to images, not continuous inputs in general.

On the quantitative analysis of deep belief networks, https://dl.acm.org/citation.cfm?id=1390266
is an older example of AIS used to evaluate generative models that could be cited.

I think it would be better to drop z_0 and T_0 in equations (1) and (2), and have z_1 sampled from p_0 directly, to make the equations consistent with the equations that follow afterwards.

"using a latent variable z with a fixed prior distribution"
In VAEs the prior can also be learned, it doesn't have to be fixed (this may in fact help alleviate the holes problem).

It could be mentioned that the objective in Eq. (9) has the same form as a generalized VAE objective, such as the one used by beta-VAE, https://openreview.net/forum?id=Sy2fzU9gl

At the bottom of page 4 and page 5, R_p(D) uses a different font for R than the rest of the paper.

Fig. 1 is too small to read on paper, I had to zoom in using the pdf in order to see it properly.

Last paragraph of section 4 specifically mentions images, even though it doesn't need to (datapoints don't have to be images).

The last two paragraphs of page 7 contain a few grammatical mistakes:
- fixing the architecture of neural network --> fixing the architecture of the neural network
- as the result --> as a result
- there exist a rate distortion code for any rate distortion pairs  -->  there exists a rate distortion code for any rate distortion pair
- While in our definition of rate distortion -->Whereas, in out definition of rate distortion

Top of page 8, use \citet instead of \citep where appropriate.

Capitalize names and acronyms in references, such as ELBO, GAN, MMD, VAE, Bayes, Monte Carlo, etc.

**Experience Assessment:**

I have published one or two papers in this area.

**Review Assessment: Checking Correctness Of Derivations And Theory:**

I assessed the sensibility of the derivations and theory.

**Review Assessment: Checking Correctness Of Experiments:**

I assessed the sensibility of the experiments.

**Review Assessment: Thoroughness In Paper Reading:**

I read the paper thoroughly.

---

> ### Author Response · Authors · 2019-11-15
> **Response to the review**
>
> We thank the reviewer for the feedback.
> Reviewer: "Since this bias asymptotically goes to zero, I suspect that R^{AIS}_p may become equal to R_p for M -> infinity, and perhaps the bound improves monotonically as M increases."
>
> That is correct.
> ==============
> Reviewer: "If my understanding is correct, the reason for the inequality in proposition 4 is that log\hat{Z} is a biased estimate of logZ (due to Jensen's inequality) despite \hat{Z} being an unbiased estimate of Z. If that's all there is to it, the proof in the appendix, although precise, is a bit of an overkill."
>
> The proof of Proposition 4 is more involved than the direct use of the Jensen’s inequality of $E[\log \hat{Z}_{k}^{AIS}(x)] \leq \log Z_{k}(x)$ on Eq. 15. Applying this Jensen's inequality on Eq. 15 would result in a relationship between the rate element of two different rate-distortion pairs that do not have the same distortion element. Therefore, no conclusion can be made about the relationship between the two rate-distortion curves.
> ==============
> Reviewer: "Section 6 attributes the performance drop of VAEs in the low-rate regime to the "holes problem". If that's true, then I would expect the situation to be improved with more flexible prior / posterior models. What prior / posterior models were used in the experiments? If only diagonal Gaussians were used, then it would be interesting to see whether more flexible priors / posteriors such as normalizing flows would change the results."
>
> In the VAE experiment, we have used the standard factorized Gaussian distribution as the variational posterior. We agree with the reviewer that training a VAE with a more expressive posterior is an interesting experiment. It is straightforward for us to perform this experiment, and we will include it in the final manuscript.
> In the paper, we have two experiments that back up our claim about the "holes problem". The first experiment is the AAE experiment which shows that alleviating the holes problem by matching the aggregated posterior to the prior would result in a better low-rate performance compared to VAEs. The second experiment is the "Mixture Prior" experiment of Section 6.2, which shows that the low-rate performance of the model can detect the holes problem that is caused by using the mixture prior.

---

### Official Review · AnonReviewer3 · 2019-10-23
**Official Blind Review #3**

**Rating:** 3

**Review:**

This paper considers the rate-distortion tradeoffs of deep generative models such as variational autoencoders (VAEs) and generative adversarial networks (GANs).
The authors propose an annealed importance sampling (AIS) method to compute the rate-distortion curve efficiently. In experiments, the authors compare the rate-distortion curves for VAEs and GANs and discuss the properties of rate-distortion curves.

The method for computing the rate-distortion curves of deep generative models is interesting and the rate-prior distortion curve is promising as a performance measure. However, the main technical contribution of this work is the estimated AIS rate-prior distortion curve and it is based on a straight-forward application of AIS.

In fact, Sections 2 and 3 discuss already known result in literature although summarizing them in a paper is nice for readers.

Although the findings in the experiments are interesting and insightful, they are still preliminary and further investigations are desirable.

In Section 5, the authors mention the consistency of their framework with Shannon’s rate distortion theorem. This seems to be a little overstatement because the authors discuss little about the optimization of the prior p(z).



**Experience Assessment:**

I have published one or two papers in this area.

**Review Assessment: Checking Correctness Of Derivations And Theory:**

I assessed the sensibility of the derivations and theory.

**Review Assessment: Checking Correctness Of Experiments:**

I assessed the sensibility of the experiments.

**Review Assessment: Thoroughness In Paper Reading:**

I made a quick assessment of this paper.

---

> ### Author Response · Authors · 2019-11-15
> **Response to the review**
>
> We thank the reviewer for the feedback.
> Reviewer: "However, the main technical contribution of this work is the estimated AIS rate-prior distortion curve and it is based on a straight-forward application of AIS." and "In fact, Sections 2 and 3 discuss already known result in literature although summarizing them in a paper is nice for readers."
>
> Section 2 is the background section, and in Section 3, we are setting up the problem and defining the upper bound on the rate-distortion objective that we want to optimize. In this section, we discuss the properties of this bound and cite the relevant literature. The main technical contribution of our paper is Section 4, where we show how we can use the AIS to approximate the rate-distortion curves. We believe the application of AIS in the context of rate-distortion theory is novel. Specifically, we believe the theoretical bound that we proved for our algorithm (Proposition 4) is highly non-trivial. If the reviewer has any specific concern about related works, we would be happy to address them in the final paper.
> ==============
> Reviewer: "Although the findings in the experiments are interesting and insightful, they are still preliminary and further investigations are desirable."
>
> We have performed an extensive set of experiments with different models (GANs, VAEs, and AAEs), different architectures (fully-connected and convolutional), different datasets (MNIST, CIFAR-10), and different distortion metrics (pixelwise MSE and feature space MSE). We have extensively validated the correctness of our MCMC implementation by comparing it to analytical results (for linear VAE) and measuring the BDMC gap at different rates. We believe our experiments are sufficient to support our claims. If the reviewer has any specific concerns about any of the experiments, we would be happy to address them in the final paper.

---

### Official Review · AnonReviewer1 · 2019-10-30
**Official Blind Review #1**

**Rating:** 3

**Review:**

This paper presents a method for evaluating latent-variable generative models in terms of the rate-distortion curve that compares the number of bits needed to encode the representation with how well you can reconstruct an input under some distortion measure. To estimate this curve, the author’s use AIS and show how intermediate distributions in AIS can be used to bound and estimate rate and distortion. They apply their evaluation to GANs, VAEs, and AAEs trained on MNIST and CIFAR-10.

I found this paper well written, with a number of interesting technical contributions, particularly how to leverage AIS to compute R-D curves for an individual model. However, the utility and interpretation of these R-D curves for single models remains confusing to me, and there is insufficient discussion and comparison to other joint diversity/sample quality metrics proposed in the GAN literature. The compute time required for evaluation may also limit the applicability: 4-7 hours for 50 images on MNIST, and 7 days for CIFAR-10.

Major comments:
* How should we interpret rate-prior distortion for an individual model vs. rate-distortion where models are optimized for each rate? Past work in learned compression and generative models (Theis et al. 2016, Balle et al. 2016, Alemi et al., 2018) show that models must adjust their decoder (and prior) as a function of rate to be optimal in terms of distortion. For a fixed decoder, optimizing the prior may still be required to achieve low rate. Given that many of the models you compare are trained to do well at one point on the R-D curve, why does it make sense to evaluate them at other points? Additionally, you only evaluate models with deterministic decoders and many of the experimental conclusions are highly specific to this setting but not noted.
* As you focus on general distortion metrics instead of NLL alone, it'd be interesting to compare curves under different distortion measures, e.g. MS-SSIM for images or L1 vs. L2. Right now there's not much experimental novelty vs. prior work that looked at rate-distortion curves with NLL distortion and Gaussian observation models.
* It’d be useful to include experiments comparing Rate-Prior distortion curves and Rate-distortion curves where you a) optimize over the prior, b) optimize the decoder, fixing the prior, and c) optimize both the prior and decoder.
* There’s no comparison of other approaches to generate the rate-prior distortion curve. For example, you could just use an amortized inference network like in VAE w/ a flexible variational family and anneal beta over time.
* There are several related papers which should be discussed and contrasted, in particular https://arxiv.org/abs/1901.07821 which looks at rate-distortion-perception tradeoffs, and https://arxiv.org/abs/1806.00035 which presents precision-recall curves for diversity/quality metrics applied to implicit models. How do the insights gained from the rate-distortion curves relate to precision/recall and why should one be preferred over the other? https://arxiv.org/abs/1611.02163 also looked at distortion as a metric for GANs (equivalent to beta -> infinity in your framework).

Minor comments:
* Missing discussion of several related works: that presents a complexity measure for the latent space of GANs: https://arxiv.org/abs/1802.04874
* “Wasserstein distance remains difficult to approximate…” - see https://openreview.net/forum?id=HkxKH2AcFm that advocates for evaluation with Wasserstein
* Tightness of bound on simulated data (what BDMC provides) may not correspond to tightness of bound on real data (what you care about in practice).
* The treatment of VAEs as implicit models only makes sense with location scale family p(x|z), thus the entire framework proposed here doesn’t make sense with e.g. autoregressive p(x|z), as used in PixelVAE and others.
* Why focus on fixed prior p(z)? An alternative would be to optimize p(z), q(z|x) and fix p(x|z). How would this change the resulting rate-prior distortion curves?
* “We can compute the R-D curve by sweeping over \beta rather than by sweeping over D” - this is not the case when the R-D curve has linear segments, see e.g. Rezende & Viola 2018
* Many of the properties and discussion around rate-prior distortion functions (especially w/NLL distortion) are also in Alemi et al. 2018 as their definition of “rate” is identical to your definition of “rate-prior”. Also many of these properties are specific to continuous latents which isn’t noted.
* Should clarify that q_k(z|x) correspond to points along R_p(D)
* The results in Eqn 14-17 showing you can tractably estimate distortion and get an upper bound on rate using the AIS-derived distributions are very cool!
* “AIS variance is proportional to 1/MK” - this is for variance in the partition function? How does this translate to variance in estimates of rate/distortion?
* “In the case of probabilistic decoders, …” -> need the caveat this is with NLL distortion
* Validation on the linear VAE is great!It looks like some of the points for AIS at low distortion are below the analytic rate, but the proofs indicate the estimated rate should be greater than the analytic rate. Is this just noise?
* Fig 4 and 5: hard to see difference between the dashed lines
* VAE results would change drastically if you targeted them to different regimes (e.g. beta-VAE or constrained optimization like GECO)
* Statements like “VAE is trained with the ELBO objective, which encourages good reconstructions” only make sense when the decoder is location-scale. VAEs w/rich autoregressive decoders typically do a horrible job reconstructing.
* How robust are model differences across random initialization? It’d be great to add error bars to all these plots, especially given that GAN training can stochastically succeed.
* Eqn 18/Fig6a: depending on the dataset, you could easily notice the difference of 4.6 nats and log-likelihood could still tell these two models apart. It’d be useful to add a line at beta=1 to show that the likelihood would be the same but the R-D curves are different.

========================
Update after rebuttal

Thank you to the authors for addressing a number of my concerns, adding additional text and experiments. However, my main concern remains: if rate-distortion in an individual model is a useful method for evaluating generative models, you should compare it empirically with other metrics that have been proposed for this purpose (e.g. precision-recall). Additionally, while the theoretical novelty of getting the full R-D curve from a single AIS run is very cool, I'm skeptical of the practical utility as a metric for generative models due to the computational costs of AIS (4-7 hours for 50 images on MNIST). The simple baseline I suggested of training an amortized inference network with beta annealed over time would not require training a separate encoder for each point in R-D, you could just start beta large, and anneal beta in steps to 0 over time, tracing out an R-D curve. Given the current experiments, it's not obvious if the win of AIS in terms of accurate posterior inference is worth the increased computational cost over a simple VI baseline.

**Experience Assessment:**

I have published in this field for several years.

**Review Assessment: Checking Correctness Of Derivations And Theory:**

I carefully checked the derivations and theory.

**Review Assessment: Checking Correctness Of Experiments:**

I carefully checked the experiments.

**Review Assessment: Thoroughness In Paper Reading:**

I read the paper thoroughly.

---

> ### Author Response · Authors · 2019-11-15
> **Response to the review**
>
> We thank the reviewer for the feedback.
> The reviewer is mainly concerned about discussions with the related works and suggests new experiments. We re-wrote the related work section of the paper (Section 5 and Appendix A) and thoroughly discussed all the references pointed out by the reviewer. We also performed some of the experiments that the reviewer suggested and included them in the revised paper. We will include the remaining experiments in the final paper.
> ==============
> The reviewer is concerned about the relationship of our work with learned compression methods such as Theis et al. and Balle et al. and asks "How should we interpret rate-prior distortion for an individual model vs. rate-distortion where models are optimized for each rate?"
>
> Our work follows a conceptually different goal than Theis et al. or Balle et al. The goal of these works is to achieve the best compression rate for a given distortion and thus they can learn the decoder to minimize the compression rate. However, our goal is to evaluate a particular generative model with a fixed prior and decoder, independent of how the model was trained or the objective used during the training. We discuss the failure modes of scalar log-likelihoods and show that our rate-distortion curves provide a much richer picture of the performance of a given generative model, such as detecting the mode-missing and mode-inventing behavior of the generative model. As the reviewer points out, our rate-distortion curves are in spirit similar to the precision-recall curves. Similar to our framework, in the precision-recall setting, the decoder is not optimized to achieve a better precision for a given recall, but rather the precision-recall curves are plotted for evaluating a given generative model with a fixed decoder.
> ==============
> Reviewer: "it'd be interesting to compare curves under different distortion measures, e.g. MS-SSIM for images or L1 vs. L2."
>
> Thanks for suggesting this experiment! We performed this experiment and included the results in Section 6.3 and Figure 6 of the paper. In this section, we plotted the RD curves of GANs, VAEs, and AAEs, using the MSE on the deep features of a CNN as the distortion metric. In all cases, the qualitative behavior of the RD curves with this distortion metric closely matches the qualitative behaviors of pixelwise MSE, indicating that the results of our analysis are not overly sensitive to the particular choice of distortion metric.
> ==============
> The reviewer is concerned about the discussion of our work with related works such as perception-distortion tradeoff and precision-recall tradeoff.
>
> Thanks for pointing us to these related works! We re-wrote the related work section of the paper (Section 5 and Appendix A) and thoroughly discussed the connections of our work with the distortion-perception tradeoff and the precision-recall tradeoff. We also included a discussion about the "Unrolled GANs" experiment and the GILBO metric.
> ==============
> Reviewer: "For example, you could just use an amortized inference network like in VAE w/ a flexible variational family and anneal beta over time."
>
> Thanks for suggesting this experiment! It is straightforward for us to perform this experiment, and we will include it in the final manuscript. We wanted to point out that in the suggested experiment, we would need to train a separate variational encoder network for every point on the rate-distortion curve, whereas, by using the AIS posterior, not only we are not restricted to a variational family, but also we can plot the whole rate-distortion curve with one single run of AIS.
> ==============
> Reviewer: "The entire framework proposed here doesn’t make sense with e.g. autoregressive p(x|z), as used in PixelVAE and others."
>
> It is straightforward to apply our method to latent variable models with autoregressive decoders such as PixelVAE. In this case, we can simply use the negative log-likelihood of the PixelVAE as the reconstruction cost. We believe this is an important future direction of our work.
> ==============
> Reviewer: "How robust are model differences across random initialization? It’d be great to add error bars to all these plots."
>
> In Appendix D and Figure 8 of the paper, we have discussed the effect of the random initialization, and have shown that AIS is robust against the choice of random seed in our setting.
> ==============
> Reviewer: "Validation on the linear VAE is great! It looks like some of the points for AIS at low distortion are below the analytic rate, but the proofs indicate the estimated rate should be greater than the analytic rate. Is this just noise?"
>
> The estimated rate is an upper bound *in expectation*, which implies that it is an upper-bound averaged over different runs of AIS. This figure shows one single run of AIS (with 40 independent chains).

---

### Author Response · Authors · 2019-11-15
**New revision uploaded: more experiments/plots and extended related works**

We thank all the reviewers for the feedback. Based on the reviewers’ feedback, we have uploaded a new revision of the paper with the following changes:

* We performed a set of experiments to study the effect of perceptual distortion metrics in our rate-distortion framework. The results are included in Section 6.3 and Figure 6.
* We extended the related work section of the paper (Section 5 and Appendix A) and thoroughly discussed all the related works that were pointed out by the reviewers.

We hope these revisions address the concerns of the reviewers.

---

### Decision · Program_Chairs · 2019-12-19

**Decision:**

Reject

**Comment:**

The paper proposed a method to evaluate latent variable based generative models by estimating the compression in the latents (rate) and the distortion in the resulting reconstructions. While reviewers have clearly appreciated the theoretical novelty in using AIS to get an upper bound on the rate, there are concerns on missing empirical comparison with other related metrics (precision-recall) and limited practical applicability of the method due to large computational cost. Authors should consider comparing with PR metric and discuss some directions that can make the method practically as relevant as other related metrics.